# Arctic regional changes revealed by clustering of sea-ice observations

Amélie Simon[1,2], Pierre Tandeo[1,3], Florian Sévellec[2,3], Camille Lique[2]

[1] IMT Atlantique, Lab-STICC, UMR CNRS 6285, 29238, Brest, France

[2] Univ Brest CNRS Ifremer IRD, Laboratoire d'Océanographie Physique et Spatiale (LOPS), Brest, France

[3] ODYSSEY Team-Project, INRIA CNRS, Brest, France

**Correspondence**: Amélie Simon (amelie.simon@ifremer.fr)

# Abstract

Understanding the evolution of Arctic sea-ice is crucial due to its climatic and socio-economic impacts. Usual descriptors (e.g., sea-ice extent, Marginal Ice Zone, sea-ice age, and ice-free duration) quantify changes but do not account for the full seasonal cycle. Here, using satellite observations of sea-ice concentration (SIC) over 1979-2023, we perform a k-means clustering of the Arctic sea-ice seasonal cycle, initializing with equal quantile separation and using Mahalanobis distance. Without providing prior information, this data-driven method shows that the Arctic is best described by four types of seasonal cycles: open-ocean (no ice year-round), permanent sea-ice (full coverage with a minimum of 70% SIC), and two clusters showing ice-free conditions (SIC < 0.15), namely partial and full winter freezing. The latter has larger SIC in winter, more abrupt melting and freezing periods, and a shorter ice-free season than the former. This reduction of dimension in the data suggests that the first date of retreat is a good indicator for ice-free conditions the following summer and the first date of advance a good indicator for fully ice cover conditions the following winter. We introduce the probability to belong to each four seasonal cycles as a descriptor to monitor Arctic sea-ice changes. The pan-Arctic probability to belong to the permanent sea-ice seasonal cycle has decreased by 3.1 %/decade which is compensated with an increase of probability to belong to the open-ocean cluster (1.6 % per decade), the full winter freezing cluster (1.1 % per decade) and the partial winter-freezing cluster (0.5 % per decade). Regionally, the permanent sea-ice retraction from the Pacific side is compensated by the full winter-freezing cluster while the open-ocean cluster expansion in the Atlantic side is lost by the partial winter-freezing cluster. The new classes of partial and full winter freezing are helpful for sea ice process understanding as it refines the classical MIZ category into two distinct sea-ice clusters. The trend is primarily controlled by the tendency of the more abrupt melting and growth seasonal cycle (full winter-freezing cluster) compared to the trend of the quasi-sinusoidal sea-ice seasonal cycle (partial winter-freezing cluster). Also, from the Beaufort to the Kara Seas, the southern parts have stabilized (experiencing a new typical seasonal cycle, corresponding to the full winter-freezing cluster) and the northern part have destabilized (losing their typical permanent sea-ice seasonal cycle). Therefore, this

work provides a new way to describe Arctic regional changes using a statistical framework based on physical behaviours of sea-ice. Our study calls for a more latitudinal vision of the Arctic regions.

# Short summary

Through a machine learning technique based on seasonal cycles of sea-ice concentration from satellite data over the last four decades, our research shows that four regions are sufficient to best regionalize the Arctic. These regions are mainly organized into latitudinal bands and evolve in time and space. The descriptor proposed to monitor Arctic sea-ice changes is the probability to belong to each region. The probability to belong to the permanent sea-ice regions has decreased by 3.1 % /decade.

# Keywords

Arctic sea-ice, seasonal cycle, machine learning, clustering, climate change, satellite dataset, regionalization

# Introduction

The Arctic region has experienced rapid changes over recent decades that are expected to intensify in the future (Shu et al., 2022). For a global warming of 1°C, the Arctic has warmed by about 2.5 °C. In a 4°C warmer world, the Arctic is projected to be from 7°C to 10°C warmer (IPCC, 2021; their Figure SPM.5). One of the main mechanisms behind this Arctic amplification is the retreat of sea-ice, giving way to an open-ocean that captures more solar radiation, an effect called surface albedo feedback (Pithan and Mauritsen, 2014; Goosse et al., 2018). The observed Arctic sea-ice loss has been attributed to human influence primarily because of greenhouse gas emissions dominated by carbon dioxide and methane (Eyring et al., 2021 in IPCC, their section 3.4.1.1).

The decline of the Arctic sea-ice has profound implications for the regional environment and for almost four million people living beyond the Arctic circle. Reduced ice cover increases light availability, which can enhance phytoplankton blooms (Vancoppenolle et al., 2013). This, in turn, reshapes the food web structure (Ardyna and Arrigo, 2020) and has significant consequences for fisheries, potentially impacting catch levels and spatial distribution (Stock et al., 2017). The formation and melting of sea ice also largely influences nearly all aspects of life for marine mammals in the Arctic. A delay in winter sea-ice formation can trigger marine mammals' unusual mortality events, as it has been the case in 2018 in the Bering Sea (Siddon et al., 2020). Indigenous hunting opportunities that are dependent on the presence of sea-ice have decreased and shifted in time (Huntington et al., 2017). Besides, new ice-free regions could open industrial shipping routes and offshore oil and gas exploration with associated risks of oil spills, marine mammal strikes and noise pollution and lead to tension between nations (Galley et al., 2013; Huntington et al., 2020).

The sea-ice retreat not only affects the Arctic locally but also plays a pivotal role in the global Earth's radiative budget (Forster et al., 2021 in IPCC, their section 7.4.2.3) and a potential role in the modulation of remote large-scale oceanic and atmospheric circulation, known as Arctic teleconnections (Deser et al., 2015; Cohen et al., 2020; Simon et al., 2021; Chripko et al., 2021; Smith et al., 2022: Cvijanovic et al., 2025). Therefore, describing the evolution of the Arctic sea ice on a dynamic basis is important due to its fast evolution, which has implications for both local and global climate and socio-economic systems.

Different methods have been classically used in the literature to describe the recent changes in Arctic sea-ice. Most of them are based on the analysis of sea-ice concentration (SIC), which is obtained from satellite measurements since 1979 over the full Arctic region. In comparison, observational datasets of sea-ice thickness are available only for less than two decades and are still associated with large uncertainties (Ricker et al. 2017). The sea-ice area (SIA; integral sum of the product of SIC and area of all grid cells) or the sea-ice extent (SIE; integral sum of the areas of all grid cells with at least 15% ice concentration) enable to highlight years with exceptionally low September sea-ice cover, such as 2012 and to a smaller extent 2007,

2016 and 2020 (Parkinson and Comiso, 2013; Petty et al., 2018; Gulev et al., 2021 in IPCC, their Figure 2.20; Bushuk et al., 2024) or quantify long-term trends. For instance, the September SIE exhibits a decreasing trend of −12.7 ± 2.0 % per decade over the period 1979 to 2020 (Meier and Stroeve, 2022). However, trends of SIA or SIE only inform about changes in regime from ice to open-ocean and do not consider changes in sea-ice features.

Other diagnostics have been proposed to document changes in sea-ice features. A classification using thresholds of 0.15 and 0.8 SIC is commonly used and these regions refer to the Marginal Ice Zone (MIZ). Rolph et al., (2020) noted that the MIZ shifted northward and its extent remained relatively constant. Song et al., (2025) quantify this northward shift of the MIZ of approximately 0.051°/yr. They also provide insights into the evolution of the MIZ regions using two morphological parameters (shape and compactness indices). They show that in late summer, MIZ evolves to a smoother and more compact ice edge. The thresholds are convenient to represent a category with loose and packed ice but somehow arbitrary and other definitions of the MIZ have been proposed in the literature based on dynamical considerations (e.g. Sutherland and Dumont 2018). Here, we quantify directly the regions without arbitrary threshold nor intermediate integrated metrics.

Also, the age of sea-ice categorizes sea-ice into three types: open-water, first-year ice and multi-year ice (Kwok et al., 2007; Regan et al., 2022). Maslanik et al. (2011) show a strong decrease in the proportion of multiyear ice in the Arctic Ocean during the 1980-2011 period, especially in the Canadian sector. Another diagnostic deals with the duration of the ice-free period, and quantifies the timing of the transition between the freezing and melting seasons. The recent Arctic sea-ice reduction has resulted in a longer ice-free season (~ 5-10 days per decade), due to both earlier ice retreat and later ice advance (Stammerjohn et al., 2012; Stroeve et al., 2014; Lebrun et al., 2019), especially in the Chukchi, East Greenland and northeast Barents seas (Markus et al., 2009; Parkinson, 2014; Johnson & Eicken, 2016). However, these diagnostics do not consider the full seasonal cycle of sea-ice, and thus do not inform on the sea-ice dynamics including melting and growth behaviour.

These four ways of describing the variations in Arctic SIC (SIE, MIZ, sea-ice age, ice-free duration), without considering directly the full sea-ice seasonal cycle, have nonetheless highlighted changes in the shape of the sea-ice seasonal cycle: (i) the trend in SIE (Fox-Kemper et al., 2021 in IPCC, their Figure 9.13; Meier and Stroeve, 2022), the trend in MIZ fraction (MIZ extent divided by SIE; Rolph et al., 2020) and trend in northward shift of the MIZ (Song et al., 2025) depends on the season, being maximum in late summer (ii) Arctic sea ice has shifted to younger ice between 1979 and 2018 (IPCC, 2019) and (iii) the trend of later ice advance is expected to eventually double that of earlier retreat over this century, shifting the ice-free season into autumn (Lebrun et al., 2019). Here, in this paper, we describe the evolution of the Arctic by delimiting spatio-temporal regions having a common type of seasonal cycle.

Regionalizations of the Arctic have been proposed previously. Parkinson et al., (1978) divided the Arctic into 8 regions based on either geographical boundaries or physical criteria (e.g.; the Central Arctic encompassing the largest mass of perennial sea-ice or the Greenland Sea which allows for the only deep-water connection within the Arctic Basin). This regionalization was expanded by splitting regions into individual seas to distinguish the behaviour of the Arctic coastal regions, resulting in considering up to 15 or 18 regions (Meier et al., 2007; Peng and Meier, 2018). Besides, five climatic regions of the Arctic have been defined using multiannual averages of a number of meteorological elements computed for the first half of the 20th century: Atlantic, Siberian, Pacific, Canadian and Baffin Bay regions (Przybylak, 2002, 2007). Other regionalizations have been used to assess the influence on lower latitude climates of Arctic sea-ice loss from specific areas (5 to 7 regions; Levine et al., 2021; Delhaye et al., 2024). However, the criteria for the boundaries of these proposed regions are hard to determine and somewhat arbitrary. A statistical regionalization method based on observed SIC has been proposed for Antarctica. Raphael and Hobbs, (2014) isolates regions around Antarctica by using sea ice extent decorrelation length scale and variance. The resulting five sectors exhibit distinct times of sea-ice advance and retreat. Their methodology does not account for the temporal evolution of the sectors. The originality of our analysis resides in the fact that we regionalize the Arctic based on physical criteria of the dynamics of the sea-ice seasonal cycle, therefore without

imposing pre-defined regions and allowing the regions to evolve in time. To do so, we set up a clustering method (unsupervised machine learning).

Regionalizations determined from clustering methods have been shown to be an efficient tool. It has been applied to ocean temperature profiles to capture coherent physical changes (e.g. the water column during an El Niño event (Houghton and Wilson, 2020), heat distribution in the North Atlantic (Maze et al., 2017)) or on seasonal cycle of phytoplankton biomass to identify bioregions in the Mediterranean (d'Ortenzio and Ribera d'Alcalà, 2009). The same conceptual methodology has also been applied to the polar regions. In the Antarctic, Wachter et al. (2021) described the spatio-temporal sea-ice variability and documented significant spatial shifts during 1979-1998 and 1999-2018 by means of 10 clusters based on the seasonal cycle of sea-ice. In the Arctic, Valko (2014) proposed a regionalization based on geographic and geopolitical indicators, ending up with respectively two and three clusters, and Johannessen et al. (2016) identified 6 major regions by clustering annual average of surface air temperature. The boundaries of the defined clusters coincide with the outlines of the continents and the averaged position of the sea-ice edge. Besides, clustering methods for other purposes than regionalization have been used in the Arctic. Gregory et al., (2022) using a clustering analysis together with complex networks, show that climate models underestimate the importance of some regions (Beaufort, East Siberian, and Laptev seas) in explaining the pan-Arctic summer SIA variability. Using an ocean-sea ice general circulation model, Fuckar et al. (2016) performed a k-means cluster analysis on pan-Arctic detrended sea-ice thickness and found that the associated binary time series of cluster occurrences exhibit predominant interannual persistence with a mean timescale of about 2 years. However, no spatio-temporal regionalization based on the clustering of the Arctic seasonal cycle of sea-ice has been proposed so far.

In this paper, we determine Arctic regions based on statistically different sea-ice concentration seasonal cycles, and describe Arctic changes through the time evolving borders. We identify for the first time spatio-temporal regions of the Arctic based on the variability of the seasonal cycle of Arctic sea-ice concentration. We apply a k-means clustering method to determine regions based on their time-evolving belonging to a given type of seasonal cycle. In section 2, the dataset, domain of interest

and clustering method are detailed. In section 3, we first analyze the clustering outputs of the Arctic sea-ice seasonal cycle (3.1), then examine the probability to belong to each cluster (3.2), and finally investigate the regime stability and transition (3.3). Conclusions and discussion follow in section 4.

# 2. Data and Clustering Method

## 2.1 Sea-ice concentration (SIC)

The National Snow and Ice Data Center (NSIDC) provides gridded SIC fields on a 25 km polar stereographic projection obtained from passive microwave satellite measurements on daily temporal resolutions. We use the climate data record (CDR) product (Meier et al., 2021), which is based on the most recent approach combining the NASA team (NT; Cavalieri et al., 1984) and the bootstrap (BT; Comiso et al., 1986) algorithms. Because of the tendency of passive microwave measurements to underestimate concentration, the CDR chooses the higher concentration between the NT and BT algorithms and assigns it to each grid cell. The pole hole - the region around the North Pole where satellite measurements are unavailable - is filled from the average concentration of the circle of surrounding adjacent grid cells. The size of the pole hole has diminished over time due to advancements in satellite technology. Measurement uncertainties are highest at low SIC, where satellite signals are often influenced more by atmospheric and surface conditions—such as clouds, water vapor, melt on the ice surface, and changes in the character of the snow and ice surface—than by the actual presence of ice. We utilize daily data from January 1979 to December 2023, using linear interpolation for the few missing data and compute mean values every 5 days. The 29 February of every bissextile year is removed before computing the 5-day mean. We choose this 5-day temporal resolution as similar results are found for a daily temporal resolution whereas a monthly temporal resolution shows small differences in the spatial distribution of clusters (Figure S1). Sensibility tests suggest that 45 years of data are long enough to obtain a robust signal, as close clusters are obtained using periods of 20 years, 30 years and 40 years (Figure S2). Throughout the manuscript, sea-ice will always relate to concentration.

## 248 2.2 Studied domain

249 The study considers the ocean above 55°N. The description of the domain is 250 based on the delimitation provided by NSIDC (Meier et al., 2023) and encompasses 15 251 classically predefined regions (Figure 1). The bathymetric data is derived from the 252 GEBCO 2024 Grid (GEBCO Compilation Group, 2024).

253

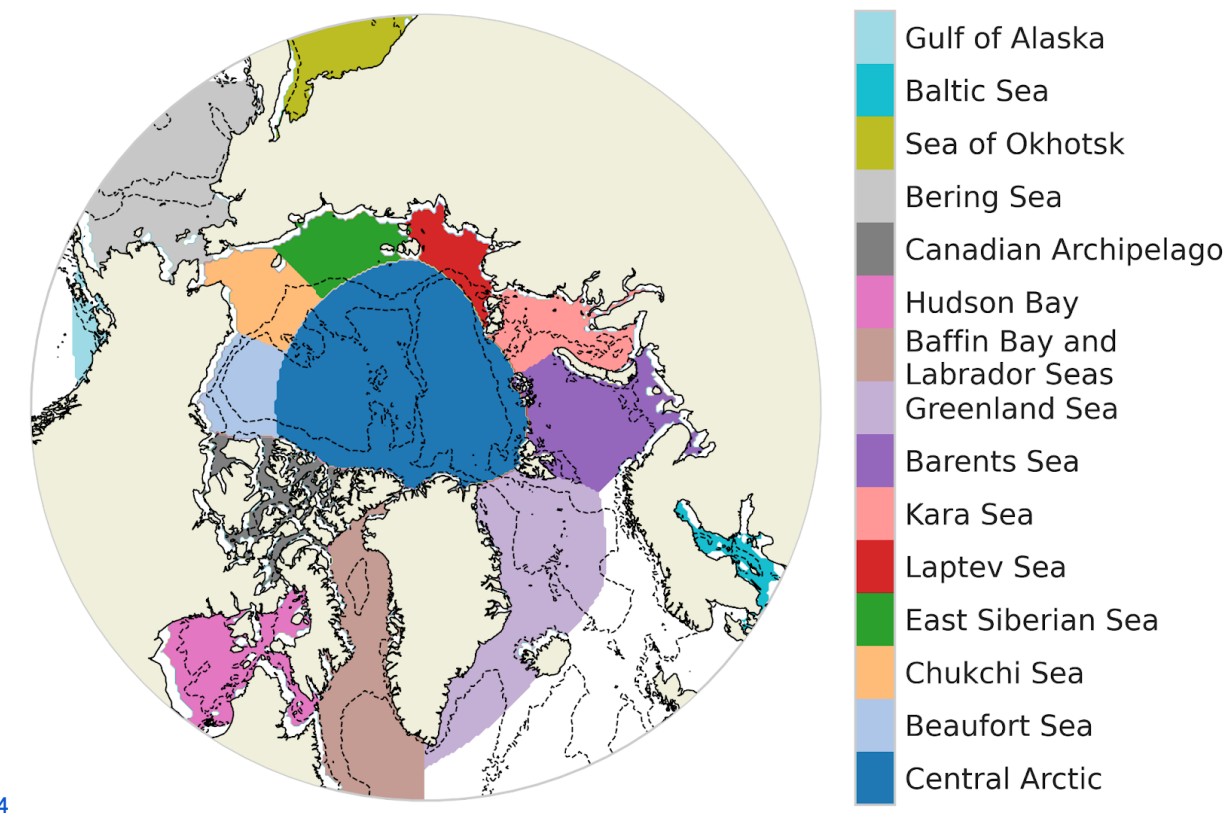

254

Figure 1: Geographical decomposition of the Arctic Ocean (defined as ocean above 256 55°N) into 15 regions following Meier et al. (2023). Bathymetry contours -100 m and 257 -2000 m are drawn with a dotted line.

## 2.3 Clustering set up

We consider all oceanic grid cells above 55°N having a non-zero sea-ice seasonal cycle (having at least a non-zero value for SIC throughout the year). Hence, the number of considered grid cells depends on the year. Grid cells with a zero sea-ice seasonal cycle are reintroduced after the clustering in order to define an open-ocean cluster. This favours a separation between regimes with and without sea-ice. The input data of our clustering are all the seasonal cycles including every considered grid cell and every year. In practice, we are thus working with a matrix with rows containing every considered grid cell of the period 1979-2023, here called points (1123710 elements) and columns containing every time step for one year, here 5-day mean (73 elements). A schematic of this matrix input data for the clustering is presented Figure 2a.

We implement a k-means clustering algorithm, which is an unsupervised machine learning method that groups data into subsamples sharing common features (Jain et al., 2010). It has the advantage of being non-parametric as our data distribution is strongly non-Gaussian. Indeed, SIC is bounded between 0 and 1 with high occurrences of values close to 0 and 1. It is an iterative method that minimizes a cost function being the sum of the squared distance (distance in a sense that would be defined later) between each seasonal cycle and its nearest cluster center (also called centroid). At each iteration, the coordinates of the centroids are updated. The initialization of centroids coordinates using k-means++ concept (the first centroid is chosen randomly, the second is the farthest-away, the third the farthest-away of the first and second, and so on) has been tested and is partly impacting our results. Therefore, we choose a different initialization strategy. We initialize the centroids coordinates based on seasonal cycles that separate the data into equal quantiles. For a clustering involving two clusters, the initializations are the two seasonal cycles of 0.33 and 0.66 quantiles of all seasonal cycles; for a clustering involving three clusters, the initializations are the three seasonal cycles of 0.25, 0.5, and 0.75 quantiles, and so on (Figure 3b). The quantiles are calculated over all the seasonal cycles considered in this study. This favours initial centroids far from each other to avoid iterating over a local minimum and the clustering is thus deterministic (i.e., it does not present any random

aspect). The strategy of initialization based on quantiles has been investigated for synthetic and real dataset and has shown a faster convergence compared to random and Kmeans++ initialization techniques (Jambudi and Gandhi, 2022).

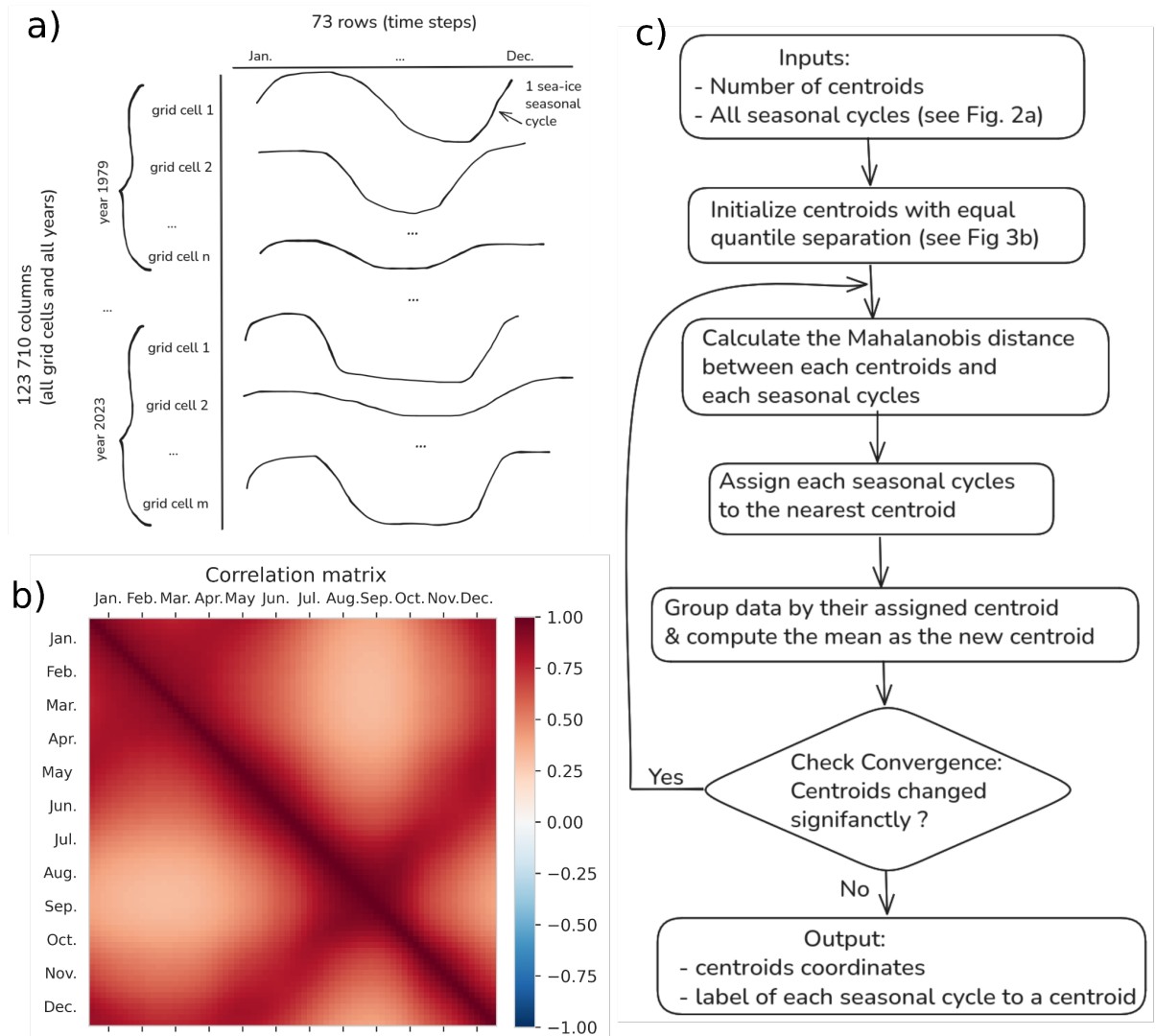

Figure 2: Schematic of the matrix input data for the k-means clustering (panel a), correlation matrix of the 5-day mean SIC for all non-zero sea-ice seasonal cycle above 55°N (panel b) and algorithm flow chart of the clustering (panel c)

The clustering algorithm is based on the calculation of distances. The Euclidean distance is often used in similar methods, yet, here, we choose to use the Mahalanobis distance (using the correlation matrix) to constrain the clustering with physical information. All the combinations of 5-day mean SIC have a positive correlation, as shown in Figure 2b by the correlation matrix. The correlation matrix is computed for all nonzero seasonal cycles for the period 1979-2023 above 55 °N. It is calculated from

the matrix of shape (73, 1123710), having 1123710 value of SIC for 73 dates. Notably, a strong correlation exists between spring and autumn (June and November), while the weakest correlations are between summer and winter (March and September, minimum correlation is 0.31). As data are correlated, a privileged direction exists when plotting the SIC for all grid cells and all years of a given date (5-day mean) against another date. We consider this physical relation of temporal dependency by using the Mahalanobis distance (which we defined as an Euclidean scalar product normalized by the inverse of the correlation matrix) in the clustering algorithm. A 5-day mean SIC strongly correlated with another (such as spring and autumn) has a reduced distance compared with Euclidean distance. We note that, as we want to conserve the physical information of the variability intensity for each 5-day mean SIC, we do not normalize the distance by the covariance matrix (as usually done for the Mahalanobis distance) but by the correlation matrix that only takes into account relation between different 5-day mean SIC. As a result, a 5-day mean SIC with weak variability (as in winter) will have a smaller impact on the total seasonal cycle than a 5-day with larger variability (as in summer). Therefore, we do not modify the relative weight (based on the variability) of each 5-day mean SIC.

The Mahalanobis norm, deriving from a symmetric operator, effectively rotates the original physical phase space (here, date of the annual cycle) to align with the data's natural directions—linear combinations of the physical time axis. This transformation allows centroid detection in a space that reflects the intrinsic structure of the data. Therefore, using the Mahalanobis distance helps the clustering algorithm to follow the direction of the correlation and capture the elongated shapes of clusters. When calculating the probability to belong to one cluster, we do not need to work with the data's natural directions, but rather work in the original physical time space. Therefore we use Euclidean distance for the calculation of probability and the correlation-based Mahalanobis distance for the clustering. An algorithm flow chart of the clustering is displayed Figure 2c.

Clustering methods are a type of unsupervised learning technique where the number of underlying classes, called clusters, is unknown beforehand. Consequently, one must test several choices for the number of clusters, k. For each chosen value of k,

a metric must be calculated to evaluate the resulting partition. The Silhouette coefficient is a metric classically used for this purpose (Rousseeuw, 1987; Houghton and Wilson, 2020). The general idea is to measure how similar an object is to its own cluster compared to other clusters; a high Silhouette value means the object is well matched to its own cluster and poorly matched to neighboring clusters. Indeed, the larger the Silhouette coefficient is (bounded between -1 to 1), the farthest the centroids are from each other and the more grouped are the points of the same cluster. It measures the quality of the clustering when seeking for compact and well-separated clusters. Ultimately, the number of clusters that maximizes the Silhouette coefficient is the optimal choice retained for the final clustering solution. We rely on the Silhouette_sample function from the python package sklearn.metrics (Pedregosa et al., 2011), which calculates the Silhouette coefficient for every point as (b - a) / max(a, b) where $a$ is the mean intra-cluster distance and $b$ is the mean distance for each point to its neighbouring cluster (closest cluster for which the point is not being part). Each point is labelled as being in a cluster using the k-means clustering (with correlation-based Mahalanobis distance), while the distance used in the calculation of $a$ and $b$ is the Euclidean distance. We have computed the Silhouette coefficient for 18 clustering (number of clusters ranging from 2 to 20; Figure 3). As the distribution of the Silhouette coefficient is asymmetric, we sort this sensitivity test using the median. The maximum median Silhouette coefficient gives an optimal number of clusters, which is three in our case (Figure 3a). Therefore, after reintroducing the open-ocean grid cell for each year, we end up with four clusters (three optimal clusters obtained using the Silhouette coefficient for non-zero seasonal cycle of sea ice and the open-ocean cluster reintroduced manually). The code developed for this study is available for download at https://github.com/amelie-simon-pro/SIC_Clustering.

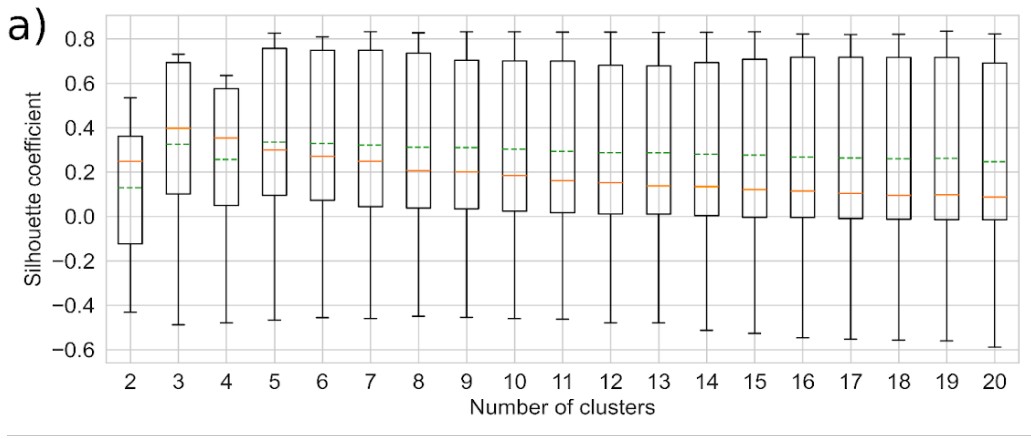

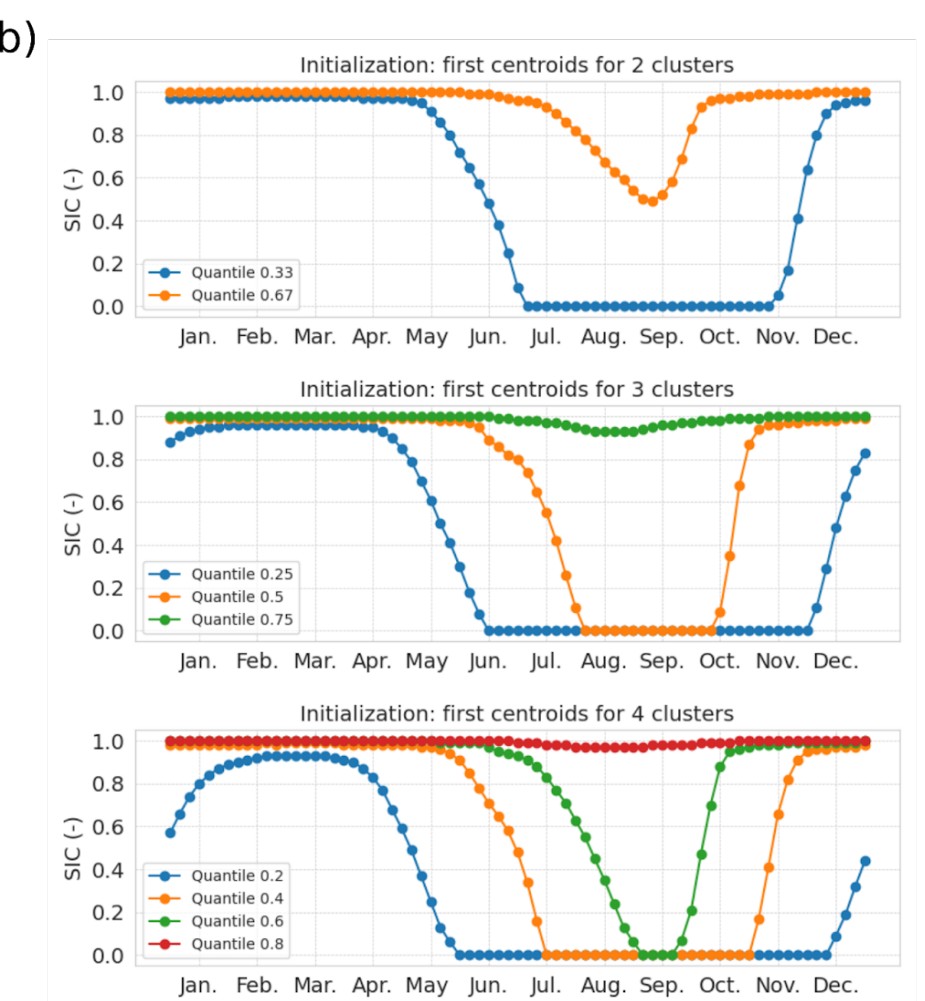

Figure 3: Boxplot of the Silhouette coefficient for a number of clusters from 2 to 20. The box extends from the first quartile (0.25) to the third quartile (0.75) of the Silhouette coefficient. The whiskers indicate the 1st and 99th percentiles. The green-dashed and orange-solid lines indicate the mean and median values, respectively (panel a). Equal quantile separation initialization: centroids of the first iteration of the clustering for a number of cluster of 2, 3 and 4 (panel b)

# 3. Results

## 3.1 Clustering outputs

The clustering method connects each seasonal cycle to a given cluster (Figure 4a) and provides the centroids of each cluster (Figure 4b). As shown in Figure 4a for year 1979 and 2023, the clustering method associates the sea-ice seasonal cycle of each year and each grid cell to the nearest seasonal cycle type (based on the smallest Mahalanobis distance between the seasonal cycle of the point and the seasonal cycle of the centroids). Without giving any information to the clustering algorithm on the spatial and temporal dependency between the seasonal cycles, we retrieve spatially continuous patterns. The clusters are sorted going toward the pole as follows: the open-ocean cluster, the partial winter-freezing cluster, the full winter-freezing cluster and the permanent sea-ice cluster. The first three clusters exhibit wavy bands surrounding the pole, and the permanent sea-ice cluster sits over the pole. More details on the description of the regions will follow based on our probabilistic framework (section 3.2).

The centroids (Figure 4b) of a cluster correspond to the average of all seasonal cycles belonging to the cluster. It is referred to as the type of seasonal cycle. They exhibit the expected physical behavior that, due to the thermal inertia of the ice and indirect processes involving the ocean and atmosphere, the maximum sea-ice coverage (in March) follows the minimum solar insolation by a lag of around 3 months, and the minimum sea-ice coverage (in September) occurs around 3 months after the maximum solar insolation (Parkinson et al. 1987).

The four types of seasonal cycles present different features. The open-ocean cluster has a SIC equal to zero all year round, which was sought for our analysis and represents year-long ice-free conditions. We refer to ice-free conditions when SIC is below 0.15. The second cluster, referred to as partial winter-freezing, has a quasi-sinusoidal shape with a mean SIC ranging from ~70% in March to no-ice in

summer (early August to mid-October). The full winter-freezing cluster is bound to a SIC of 100% from mid-November to April and to almost no-ice by mid-September. For this cluster, the sea ice completely melts in 5 months (from April to September) and totally freezes up in 2 months (from mid-September to mid-November). The full winter-freezing cluster has more abrupt seasonal changes compared to the partial winter-freezing cluster. The permanent sea-ice cluster has sea ice covered all year round, with only a partial melting between May and October, peaking at its minimum in late August (mean SIC around 70%).

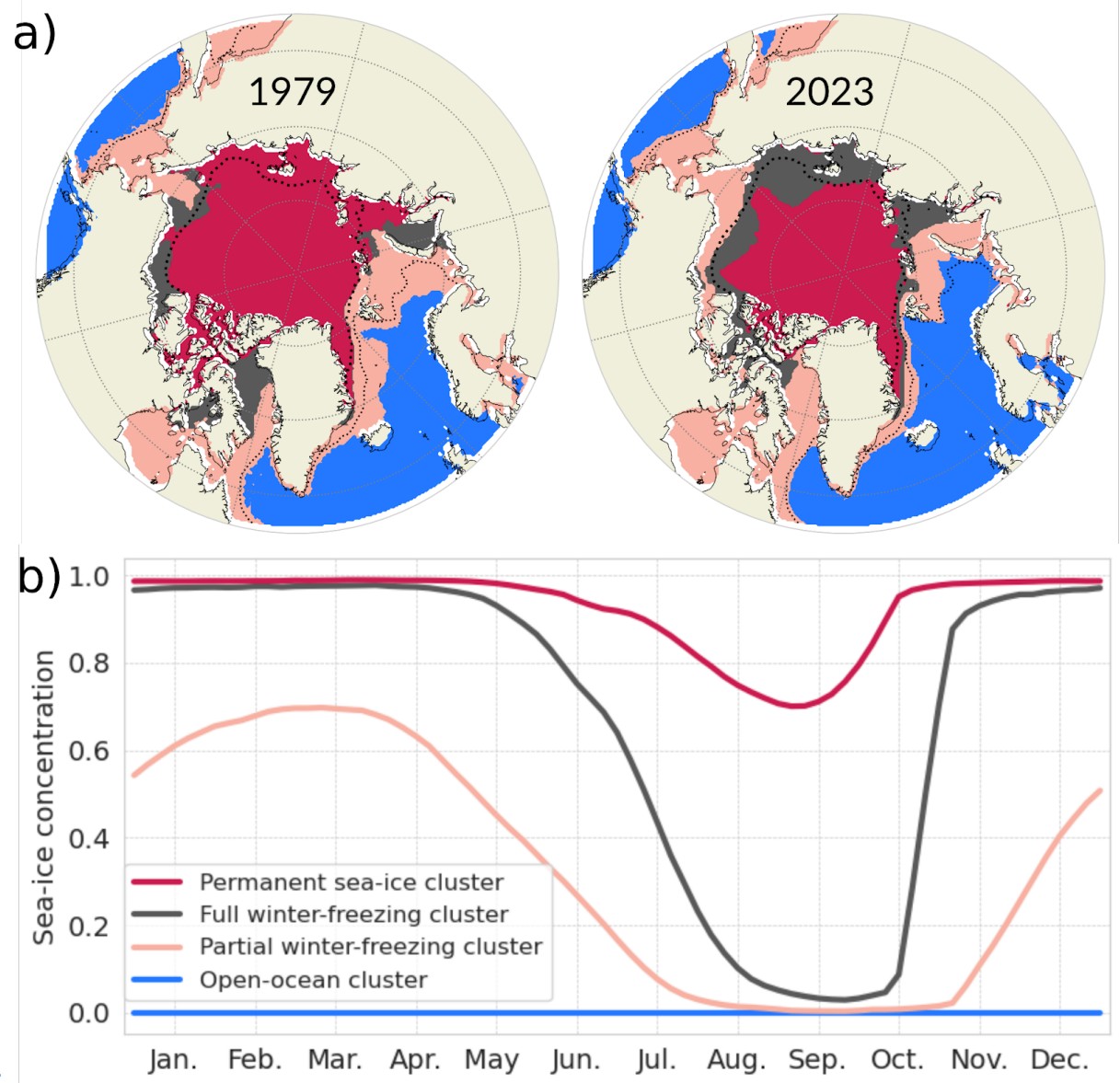

Figure 4: (a) Four types of seasonal cycles (output of the clustering method, called centroids) (b) their corresponding regions for the years 1979 (left) and 2023 (right). The dotted thin and thick lines are the mean SIC of 0.15 and 0.8 for the period 1979-2023, respectively.

This clustering analysis shed light on sea ice precursors for fully covered ice conditions and ice-free conditions, as the three clusters with sea ice have different first dates of retreat and first dates of advance. In our optimal data separation analysis, it appears that when considering areas totally covered by ice in winter (permanent and full winter freezing clusters), the first date of retreat is a good indicator for ice-free conditions in summer. Considering a given location fully ice-covered in a given winter,

our clustering results suggest that when the sea ice starts to melt in April, the seasonal cycle belongs to the full winter-freezing cluster and be ice-free the next summer. In contrast, when the melting starts one month later (in May) the seasonal cycle belongs to the permanent sea-ice cluster and the considered location will not be ice-free in summer. Besides, the freezing date for areas free of ice could differentiate between the partial winter-freezing and full winter freezing clusters and subsequently predict full ice conditions in the following winter. In our clustering, a freezing starting in October totally freezes in winter which is not the case if the freezing starts in November, having a maximum of about 70% SIC in March. Therefore, it seems that, for ice-free conditions in summer, the first date of advance is a good indicator for full ice conditions in the next winter.

However, this suggestion relies solely on the shape of the four types of seasonal cycles but to properly quantify this, the spread must be taken into account. Figure S3 displays the spread of the seasonal cycle by plotting the quantiles 0.1, 0.5 and 0.9 of each cluster. To verify our hypothesis on sea-ice indicators, we account for the spread of the date of retreat and date of advance for each cluster. To do so, we calculate the first date of retreat (the first date after the maximum SIC that is below 0.9) for each seasonal cycle experiencing fully ice covered conditions (having at least one value above 0.99 during the year). We also calculate the first date of advance (the first date after the minimum SIC that is above 0.1) for each seasonal cycle experiencing ice-free conditions (having at least one value below 0.01 during the year). For these calculations, seasonal cycles have been temporally filtered using a 15 days sliding window in order to get rid of short-term dynamical ice events, as done in Lebrun et al., (2019). To circumvent the effect of the discontinuity between 31 December and 1 January, we define the origin of time in May for the calculation of the date of advance. We then label each first date of retreat and first date of advance for each seasonal cycle with its corresponding cluster according to our clustering analysis (Figure 4a).

The normalized probability over each cluster of the first date of retreat and first date of advance at each date is shown Figure 5. This figure also displays the total number of the first date of retreat and the first date of advance of all clusters for each date. If the first date of retreat occurs between January and April, there is around 95%

of chance to belong to either the open-ocean cluster, the partial winter-freezing cluster or full winter freezing cluster, which all present ice-free duration in the following summer. However, this situation did not often occur, as the total first date of retreat happening in this period is unlikely (solely around 5% of first date of retreat for all clusters). The first date of retreat is more likely to occur between the beginning of April and August, as within this period around 90% of the total date of retreat for all clusters exist. A first date of retreat in early July has around 70% of chance to belong to the full winter-freezing cluster which present ice-free conditions in summer while a first date of retreat in early August has around 90% of chance to belong to the permanent sea-ice cluster which doesn't show ice-free conditions in summer.

The first date of advance is more likely to occur between the beginning of August until the beginning of January, as within this period around 90% of the total date of advance for all clusters exist. A first date of advance in early September has around 95% of chance to belong to the full winter freezing cluster which present fully ice covered condition in the following winter, while a first date of advance in early November has around 80% of chance to belong to the partial winter-freezing or open ocean clusters which do not show fully ice covered conditions in the following winter. Therefore, this simple model suggests that the first date of retreat could be a good indicator for ice-free conditions the following summer and the first date of advance a good indicator for fully ice cover conditions the following winter.

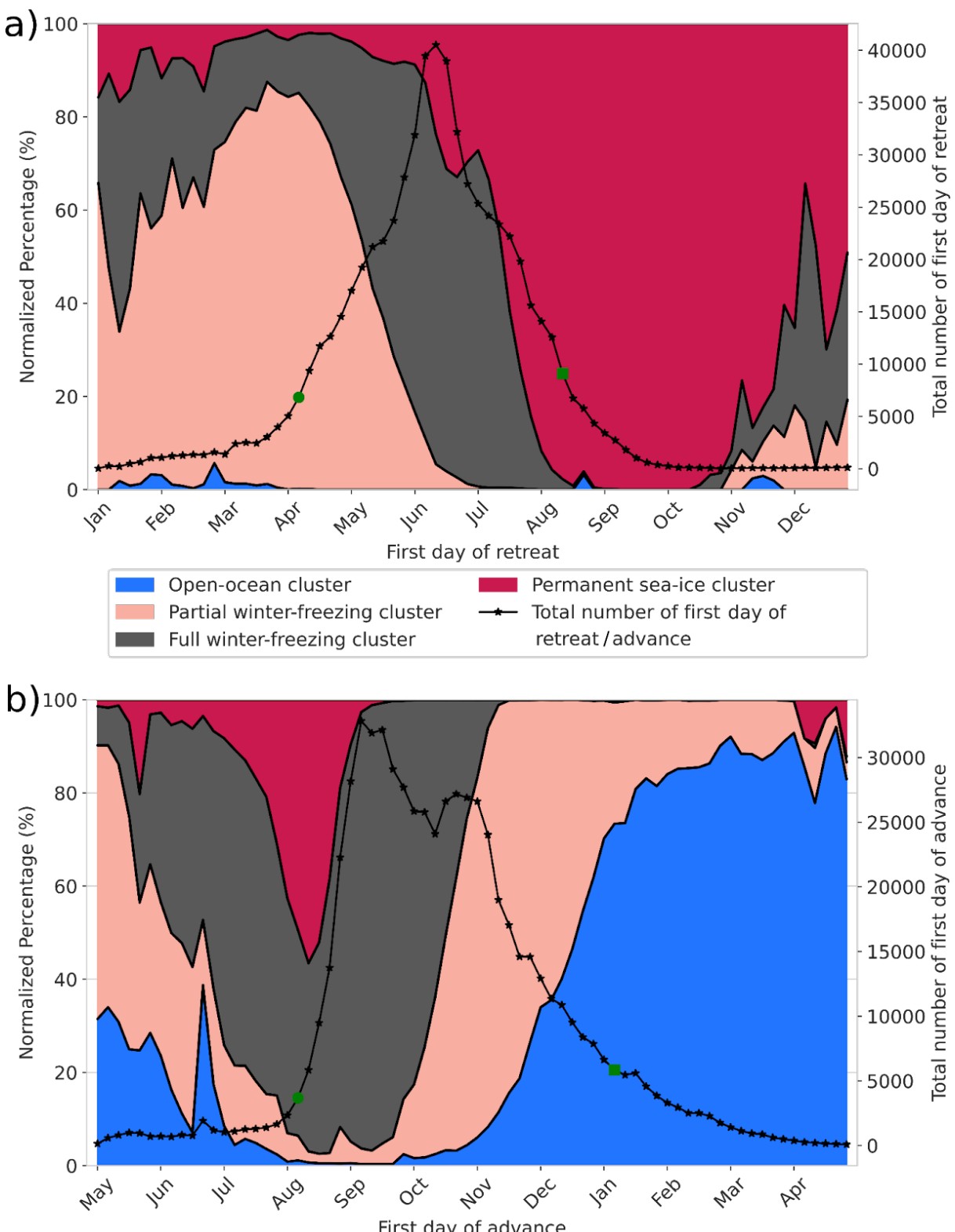

Figure 5: Normalized probability of the first date of retreat (panel a) and first date of advance (panel b) for each cluster. The solid lines with star markers are the total number of first dates of retreat and first dates of advance for all clusters. The green circle markers (start date) and green square markers (end date) cover the shortest

period where around 90% of the first date of retreat, respectively the first date of advance, for all clusters occurs.

To emphasize the added value of our clustering, we compare it to a more classical classification (Figure 6b) in which the sea ice cover is separated into the packed ice category (0.8 < SIC < 1), the Marginal Ice Zone (MIZ; 0.15 < SIC < 0.8) and the remaining, open-ocean category (SIC< 0.15; Aksenov et al. 2017). Using the cluster vision, we compute the evolution of the total area corresponding to each of our four clusters (Figure 6a).

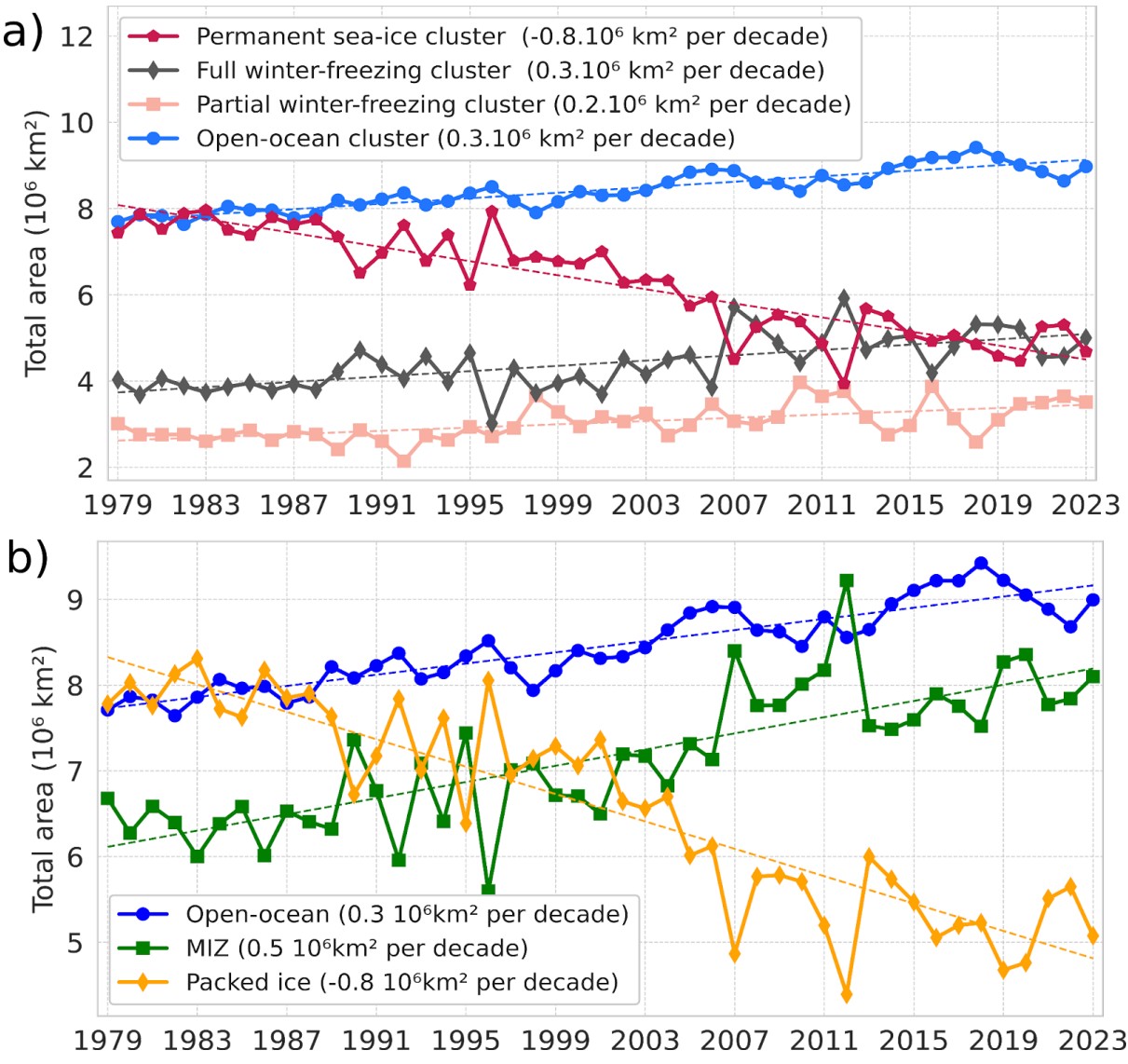

Figure 6: (a) Time series of the total area covered by each of the four clusters. (b) Times series of the area covered by three categories: packed ice (0.8 < SIC < 1), the Marginal Ice Zone (MIZ; 0.15 < SIC < 0.8) and the open-ocean (SIC< 0.15). All curves show a

significant linear trend with a p-value less than 0.05 using a Wald Test with a t-distribution.

These two methods (Figure 6a and Figure 6b) both indicate a shift toward more seasonal and ice-free conditions. Indeed, in the clustering method the permanent sea-ice cluster has notably decreased of the same amount than the packed ice in the classical categorization (-0.8 .$10^6$km² per decade). Also, the open-ocean cluster follows the same trend of the open-ocean category (0.3 $10^6$ km² per decade). The increase in the area of MIZ category is around 0.5 $10^6$ km² per decade and has been demonstrated previously (Cocetta et al., 2024; Song et al., 2025). Therefore, it appears with our clustering that the MIZ is refined into two clusters : the full winter-freezing (0.3 $10^6$ km² per decade) and the partial winter-freezing cluster (0.2 $10^6$ km² per decade). This suggests that the tendency is more likely to shift to a more abrupt melting and growth seasonal cycle (full winter-freezing cluster) compared to a quasi-sinusoidal sea-ice seasonal cycle (partial winter-freezing cluster) or, in other words, that the tendency is more likely to a total ice cover in winter with a short ice-free season (2 months, full winter-freezing cluster) than a partial ice cover in winter with a long ice-free season (4 months, partial winter freezing cluster).

Also, looking at the years with marked extremes in September sea ice extent, (2007, 2012, 2016 and 2020; see introduction), the MIZ categorization shows a transfer of area between the packed ice and the MIZ. In our clustering vision, 2007, 2012 and 2020 show a transfer of area between the permanent sea-ice cluster and full winter-freezing cluster while 2016 show a transfer of area between the full winter-freezing and the partial winter freezing, reflecting different dynamical changes in the sea-ice seasonal cycles. Therefore, our clustering analysis presents a more detailed description of the MIZ category.

As a given seasonal cycle can be in between two or more seasonal cycle centroids, we introduce the probability to belong to one cluster in the next section.

## 3.2 Probability to belong to a cluster

### 3.2.1 Calculation

To calculate the probability P of a grid point to belong to each cluster. We define the vectors **x** and **c(k)**, corresponding respectively to the SIC observed at a grid cell over a year (i.e., 73 intervals of 5 days) and the cluster centroid k. These are of dimension (73x1) and are written as:

$$\mathbf{x} = [x_1, \ldots, x_{73}]^T;$$
$$\mathbf{c(k)} = [c_1(k), \ldots, c_{73}(k)]^T \qquad (1)$$

The Euclidean distance scalar between the point x and the centroid k is defined as follows:

$$d_{x,c(k)} = \sqrt{(\mathbf{x} - \mathbf{c(k)})^T(\mathbf{x} - \mathbf{c(k)})} \qquad (2)$$

The probability P reads:

$$P(x, k) = \left[ \sum_{l=1}^{n_c} \left( \frac{d_{x,c(k)}}{d_{x,c(l)}} \right)^2 \right]^{-1} \qquad (3)$$

with $n_c$ the total number of clusters (four in our case). *P* ranges from 0 to 1 and the sum over the four clusters of *P* equals 1. In other words, the probability of being in a cluster is set by the distance of one seasonal cycle to a seasonal cycle centroid, normalized by the sum of the Euclidean distance to all clusters. This means that we use a "fuzzy" k-means clustering where the assignment is soft (each data point can be a member of multiple clusters) in contrast to a hard or crisp assignment (each data point is assigned to a single cluster; Jain et al., 2010).

We call the total probability, Pt, the normalized area weighted probability over all grid cells. We sum, for each year, the probability weighted by the area of each grid cell over all grid cells divided by the sum of the probability weighted by the area of each grid cell over all clusters and all grid cells. Pt can be written as:

$$Pt(k) = \frac{\sum_x P(x,k) \cdot \text{area}(x)}{\sum_k \sum_x P(x,k) \cdot \text{area}(x)} \qquad (4)$$

## 3.2.2 Trend of the pan-Arctic probability to belong to a cluster

After attributing each point to a probability of belonging to each cluster per year (using equation (4)), we analyze in a spatially integrated way the pan-Arctic evolution of the total probability to belong to a cluster (normalized area-weighted probability). The probability of belonging to the open-ocean cluster is around 40%, to the permanent sea-ice cluster is around 29% and to the full winter-freezing cluster is around 18 % and to the partial winter-freezing cluster is around 13% (Figure 7). Note that the absolute value reflects our choice of domain, here above 55 °N.

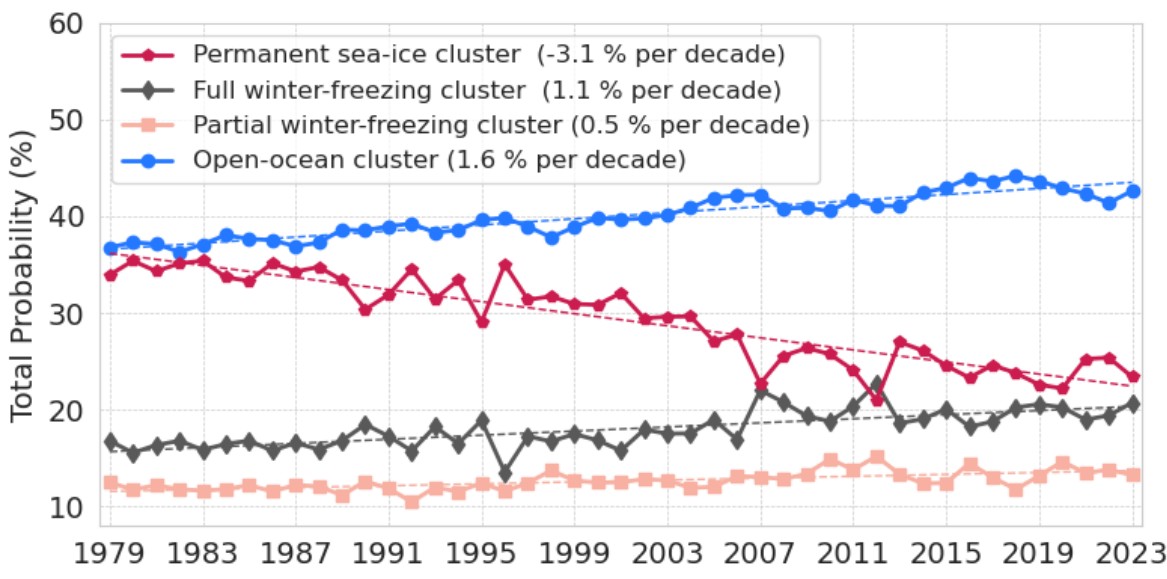

Figure 7: Evolution of the total probability (see Equation (4)) to belong to each cluster. All clusters show a significant linear trend with a p-value less than 0.05 using a Wald Test with a t-distribution

However, the time evolution of these clusters is in direct relation to the dynamics of the Arctic sea ice. A linear regression analysis indicates that the trends for all clusters are statistically significant, with a p-value less than 0.05 using a Wald Test with a t-distribution. The total probability of belonging to the permanent sea-ice cluster overall declines by around 3.1% per decade with an acceleration around the

1997-2012 period. The total probability of the three other clusters shows a decline, firstly for the open-ocean cluster (1.6% per decade) and to a smaller extent full winter-freezing (1.1%) and the partial winter-freezing (0.5% per decade). Therefore, most of the pan-Arctic probability loss over the last 45 years from the permanent sea-ice cluster is compensated by a gain of the open-ocean cluster and to a smaller extent to the full and partial winter-freezing clusters.

### 3.2.3 Regional probability to belong to a cluster

To analyze spatial redistributions of clusters over time, we average the probability (calculated equation (3)) over three periods of 15 years (Figure 8). During the first period (1979-1993), the Nordic Seas, the Bering Sea and the Gulf of Alaska belonged solely to the open-ocean cluster (free of ice). Going northward, an outer belt shape connecting the southern Barents Sea, the northern and east Greenland Sea and the southern and east Labrador Sea in the Atlantic side and the northern Bering Sea and Sea of Okhotsk mainly belongs to the partial winter-freezing cluster. Further north, an inner belt shape tight to the Arctic coast (Beaufort Sea, Chukchi Sea, East Siberian Sea, Laptev Sea, southern Kara Sea) and to the northern Barents Sea, and Baffin Bay mainly belong to the fully winter-freezing cluster. The central Arctic predominantly belongs to the permanent sea-ice cluster. The edge of the 0.3 probability of belonging to the permanent sea-ice clusters of the period 1979-1993 follows the border of the Marginal Ice Zone (0.8 SIC) located in the Central Arctic. Some regions do not have a dominant cluster but instead have a strong probability of belonging to more than one cluster, such as the northern Kara Sea, the northern Greenland Sea and the Hudson Bay.

In the subsequent periods (1994-2008 and 2009-2023), the open-ocean cluster continuously expanded in the Barents Sea, East Greenland Sea and Labrador Sea. In these same regions, the other three clusters (partial winter-freezing, full winter-freezing and permanent sea ice clusters) retract. The permanent sea-ice cluster exhibits substantial change, with intense shrinking from the Pacific side of the central Arctic, losing areas in a belt shape from the Beaufort Sea to the Laptev Sea which is

mainly gained by the full winter-freezing sea-ice cluster. This indicates increasingly frequent summer ice-free conditions during the 1979-2023 period.

Therefore, spatial redistributions in the clusters occur over time. The permanent sea-ice retraction from the Pacific side is compensated by the full winter-freezing cluster and the open-ocean cluster expansion in the Atlantic side is compensated by loss of the partial winter-freezing cluster.

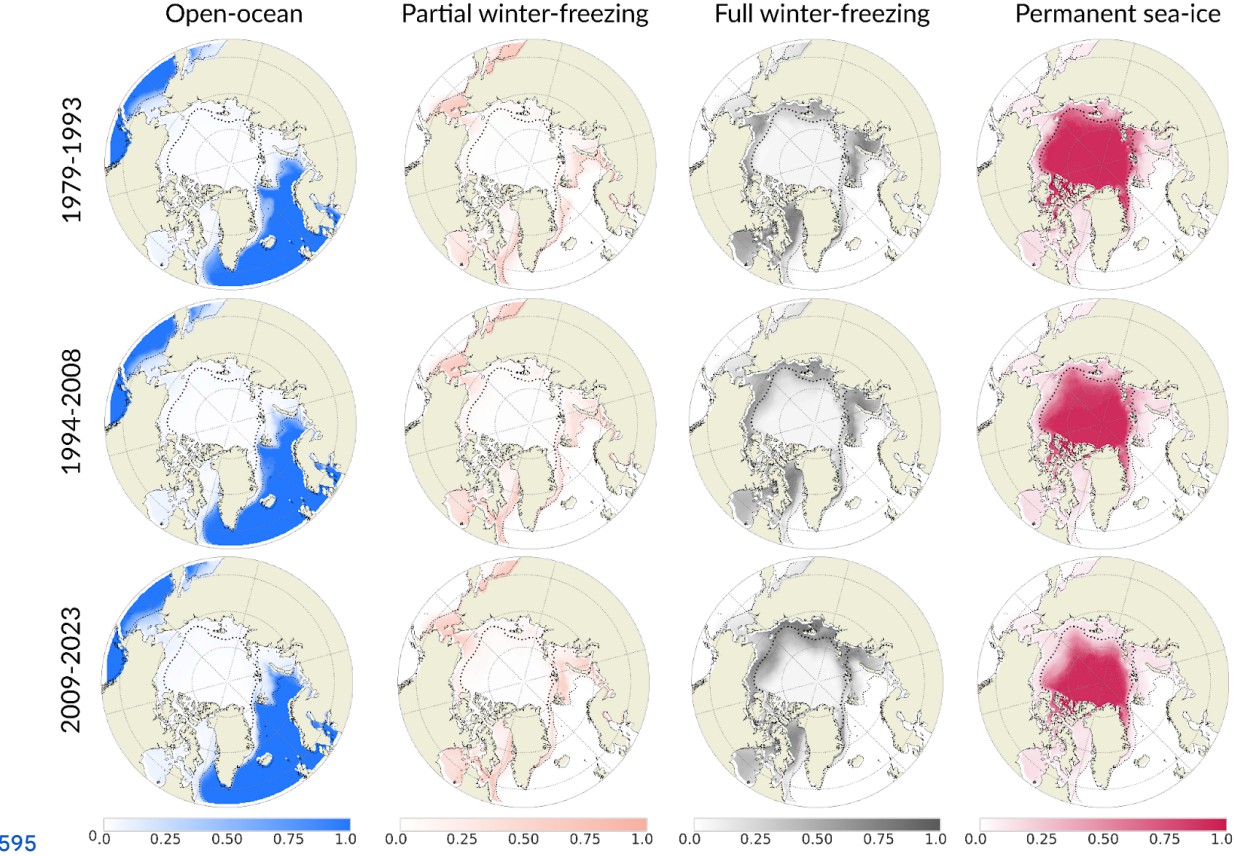

Figure 8: Map of the probability of each cluster: open-ocean (first column), partial winter-freezing (second column), full winter-freezing (third column) and permanent sea-ice (fourth column). Rows correspond to three periods of 15 years: 1979-1993 (top row), 1994-2008 (middle row) and 2009-2023 (bottom row). The dotted thin and thick lines are the mean SIC of 0.15 and 0.8 for the period 1979-2023, respectively. The circle sitting over the north pole is the pole hole (see section 2.1).

Therefore, over the whole period (1979-2023) the open-ocean cluster resides predominantly in the southern part of the Arctic and the permanent sea-ice cluster in

the central Arctic. These two clusters have no or weak seasonal changes (constant zero for open-ocean clusters and variation between 100% and 70% SIC for permanent sea-ice). To better shape our understanding of seasonal cycles which strongly change (from no ice to 70% SIC for the partial winter-freezing clusters and to 100% SIC for the full winter-freezing cluster), we distinguish which areas are mainly associated with each of these two clusters by plotting the difference of probability between these two clusters for the whole period (Figure 9). It displays spatially consistent regions. The inner belt connecting the Baffin Bay to northern Barents is attached to the coastal Arctic and is dominated by the full winter freezing cluster. Further south, this cluster is surrounded by an outer belt from the southern Barents to the southern Labrador Sea and by the Bering Sea dominated by the partial winter-freezing cluster. Thus, the full winter-freezing cluster is more likely to occur in coastal areas than the partial winter-freezing cluster. This spatial repartition might be explained by the difference in year-round shapes of the seasonal cycles: quasi-sinusoidal for partial winter-freezing and asymmetric for full winter-freezing. Indeed, Eisenman (2010) demonstrates that the coastlines, by blocking the sea-ice growth, drive the asymmetric seasonal cycle's shape while sea-ice free to grow and melt (not being blocked by land) has a sinusoidal shape. Our results corroborate this finding.

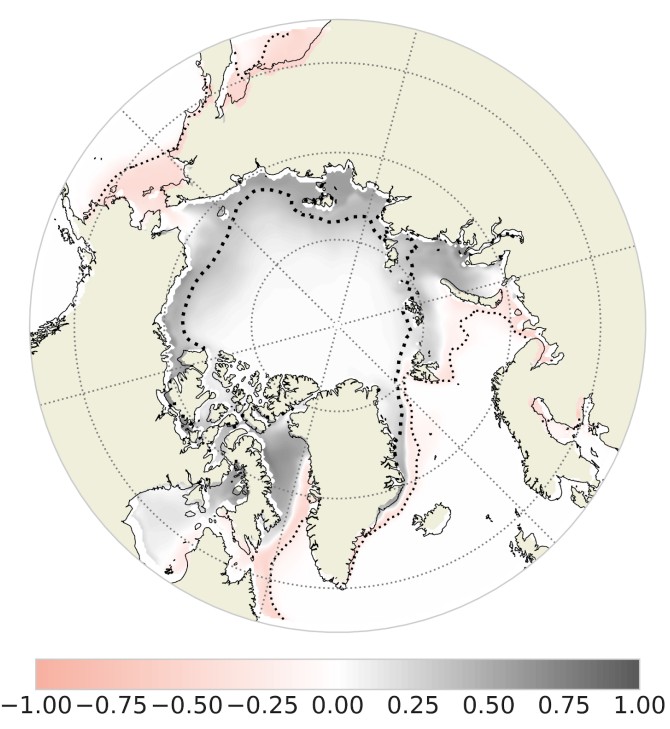

Figure 9: Map of the probability of the full winter-freezing cluster minus the partial winter-freezing cluster averaged over the period 1979-2023. The dotted thin and thick lines are the mean SIC of 0.15 and 0.8 for the period 1979-2023, respectively.

## 3.3 Regime stability and transition

In order to describe the grid-cell evolution of the Arctic sea ice over the period 1979-2023, we further classify each grid cell into four regimes: stable, unstable, destabilization, and stabilization. First, we define a stable phase as a sequence when the cluster having the maximum probability stays the same for at least 10 years in a row, allowing for a tolerance of 1 year to belong to a different cluster within that period. Sensitivity tests have been performed on this definition (Figure S4), and the results do not change when we apply small definition changes (i.e., 9 to 11 years minimum length of the same cluster with zero to 2 years of tolerance). Second, we label each grid cell as follows:

1. Grid cells being in a unique stable phase over the whole period (1979-2023) are labelled stable regime;
2. Grid cells belonging to a stable phase at the end of the period and not being in a stable phase before or being in another stable phase (with a different dominant cluster) before are labelled stabilization;
3. Grid cells not being in a stable phase at the end of the period and being in a stable phase before are labelled destabilization;
4. Grid cells not being in a stable phase during the whole period or one or several stable phases between periods of not stable phases are labelled unstable.

Figure 10 illustrates how we define the stabilization and destabilization labels.

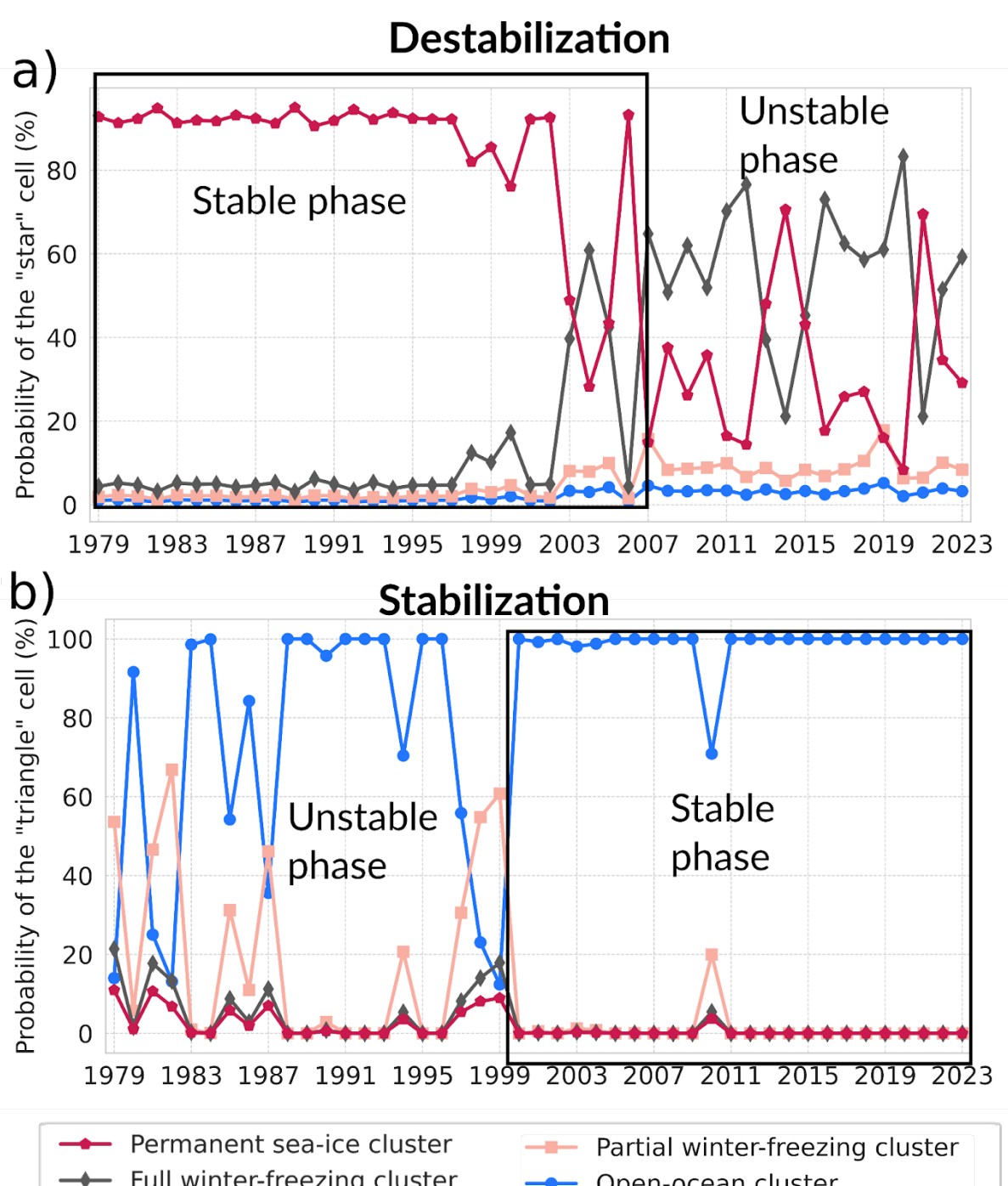

Figure 10: Evolution of clusters at the location denoted by the star (a) and the triangle

(b) in Figure 11. The stable phase is delimited by a black rectangle. These locations

have been chosen to illustrate the destabilization and stabilization label of the Arctic

sea ice evolution, respectively.

661

As shown in Figure 11, the stable region predominantly covers the central part of the Arctic Ocean, including the area around the North Pole, following most of the regions covered by permanent sea-ice cluster, as well as the ocean regions in the open-ocean cluster. Smaller regions present stable conditions: the northern Baffin Bay and southeast of Kara Sea dominated by the full winter-freezing cluster and northern Bering Sea associated with the partial winter-freezing cluster (Figure 8). Some regions jump between two or more clusters during the whole period, experiencing an unstable regime. These regions are sparse, the biggest being the northern Barents and Kara Seas. Most unstable regime areas are sitting in between stabilization and destabilization regimes areas.

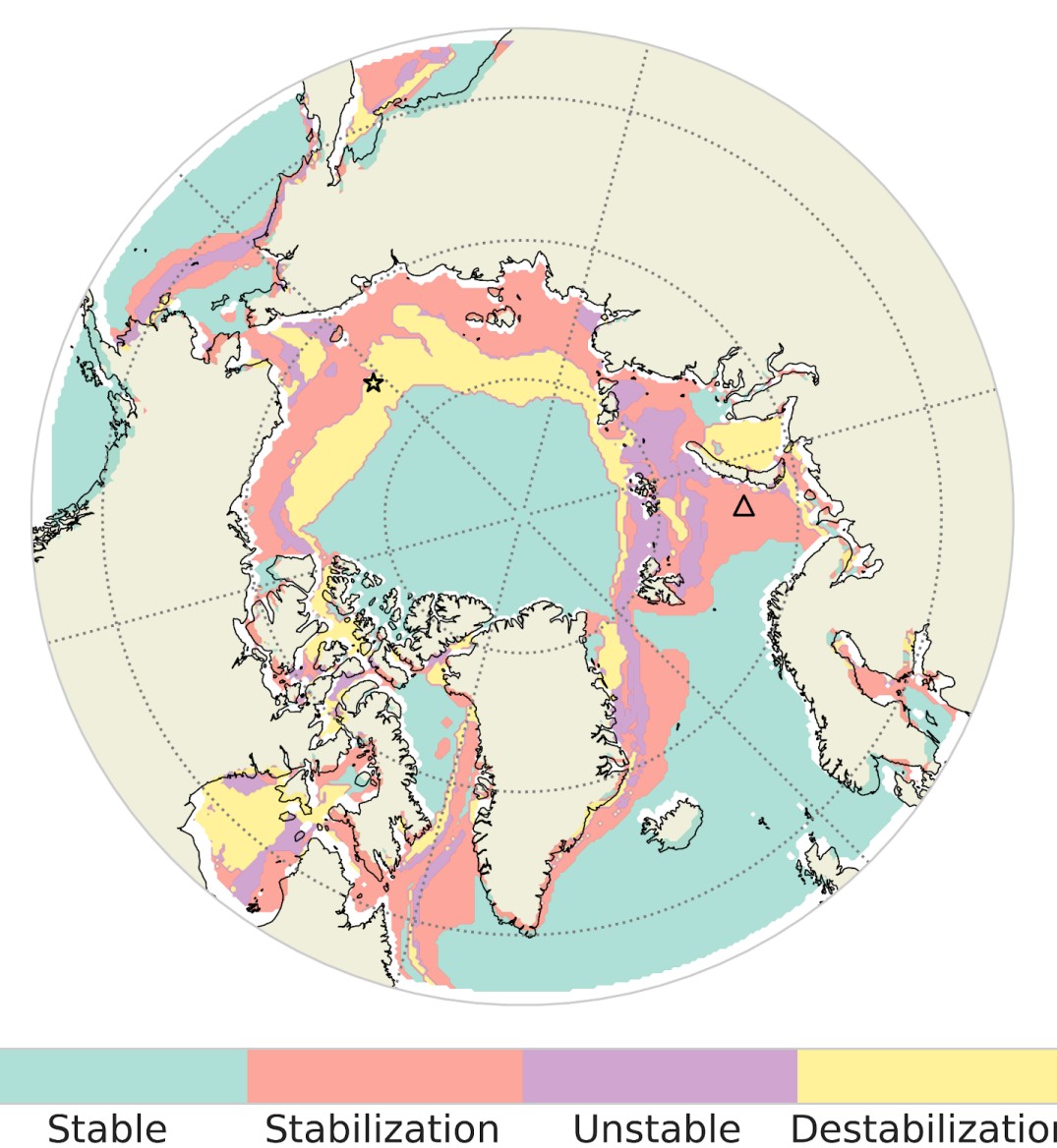

Stable    Stabilization    Unstable    Destabilization

Figure 11: Map of the four regimes (stable, stabilization, unstable, and destabilization) used to describe the evolution of Arctic clusters based on sea-ice seasonal cycles. The star and triangle markers indicated the two localizations used to illustrate the destabilization and stabilization in Figure 10 respectively.

To describe the stabilization and destabilization regimes, we display the dominant cluster (the cluster having the maximum probability) during the stable phase of these two regimes (early period for the destabilization regime and late period for the stabilization period; Figure 12c and 12d). And, to quantify the year of transition, we introduce the year of stabilization as the first year when the stable phase occurs until the end of the whole period (Fig. 12a), and the year of destabilization as the last year of the stable phase (Fig. 12b). One should note that according to our definition the maximum year of stabilization is 2013 and minimum year of destabilization is 1989.

An inner belt shape (from southern Beaufort Sea to the southern Kara) connecting parts of the Barents Sea and around Greenland (Greenland Sea and Labrador Sea) is labeled stabilization (Figure 11). The inner belt shape stabilized to the full winter-freezing cluster while the other regions in the Atlantic side (Barents to Labrador Sea) stabilized to the open-ocean cluster (Figure 12c). This is in link with the shift of the permanent sea-ice cluster to the full winter freezing cluster in the Pacific side and the expansion of the open-ocean cluster in the Atlantic side (Figure 8). This transition occurred in a northward propagation starting in the 80's in the Barents Sea for the Atlantic side and in the Laptev in for the Pacific side (Figure 12a).

The belt from the northern Canadian Archipelago to the northern Greenland Sea (wider from the Beaufort Sea to Laptev Sea) is in the destabilization regime. These regions lost their typical permanent sea-ice cluster (Figure 12d) being mainly replaced by the full winter freezing cluster (Figure 8) in a northward propagation. The southeast Kara Sea and Hudson Bay, in a northward propagation for the former and during the 2000's for the Hudson Bay (Figure 12b).

In summary, the four regimes illustrate how different regions of the Arctic have experienced changes in stability. This regionalization suggests a more latitudinal vision of the region, as for the Seas from the Beaufort to the Kara Seas, the southern parts is experiencing a stabilization to a new cluster and the northern part a destabilization of an old cluster.

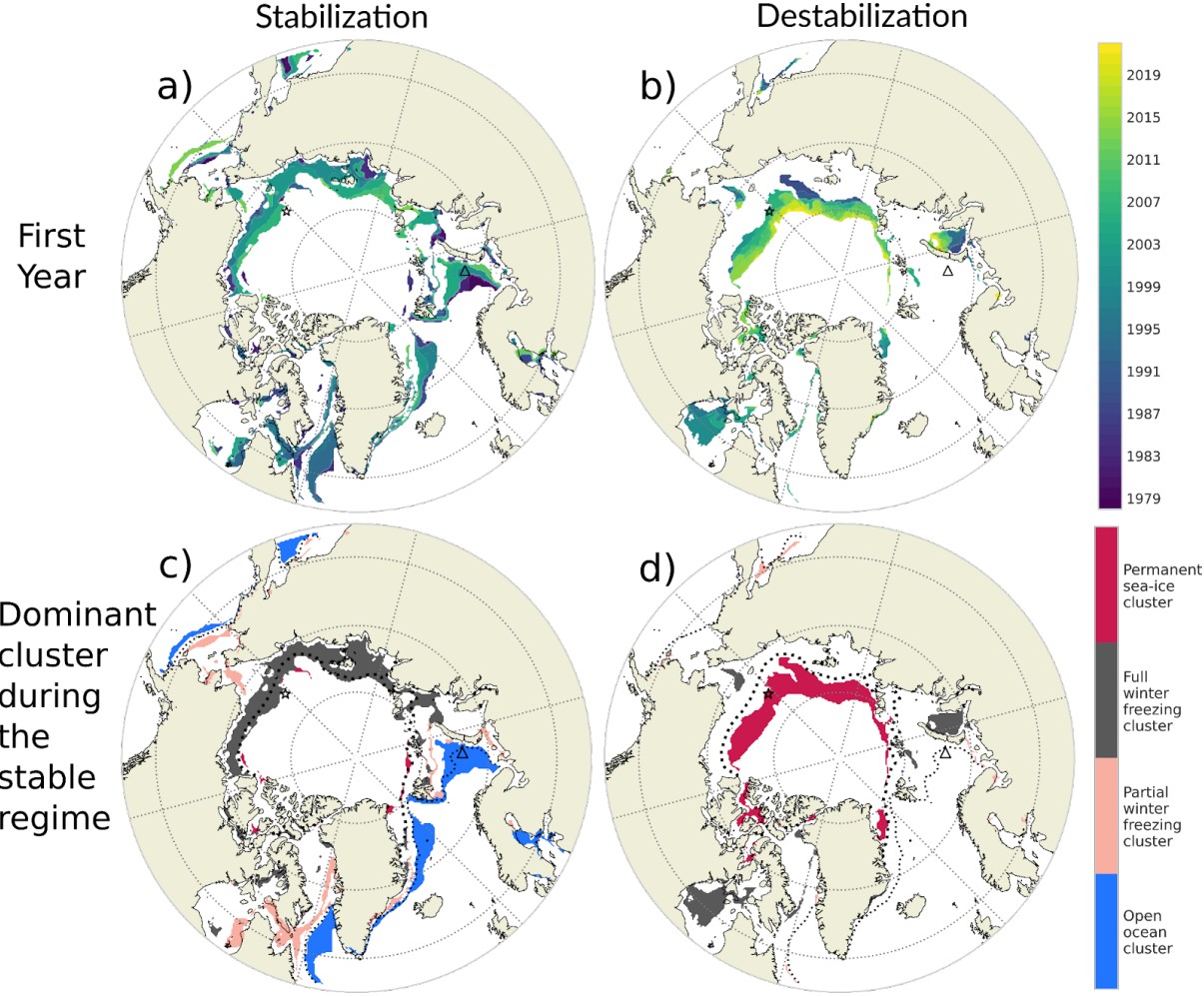

Figure 12: First year of stabilization (a) and destabilization (b) and associated dominant cluster for the stable regime of the stabilization (c) and destabilization (d). The star and triangle markers indicated the two localizations used to illustrate the destabilization and stabilization in Figure 10, respectively.

# 4. Conclusion and Discussion

This paper explores the use of a data-driven method using satellite observation of SIC to study the spatio-temporal evolution of sea ice in the Arctic over the period 1979-2023. We determine Arctic regions based on statistically different sea-ice seasonal cycles, and describe Arctic changes through the time evolving borders. The methodology is based on the clustering (machine learning method) of the full sea-ice seasonal cycle, instead of classic descriptors used in previous studies (e.g., sea-ice extent, Marginal Ice Zone, sea-ice age and ice-free duration). It shows that the Arctic sea ice changes are best described with four clusters of seasonal cycles: the open-ocean cluster (with no ice during the whole year), the permanent sea-ice cluster (total sea-ice coverage with a minimum of 70% SIC in September), and two clusters showing ice-free conditions in late summer, namely the partial winter-freezing cluster and the full winter-freezing cluster. The full winter-freezing cluster has a larger SIC in winter, displays a more abrupt summer melting and winter freezing and has a shorter ice-free season than the partial winter-freezing one. The central Arctic belongs to the permanent sea-ice cluster. According to this clustering, a first date of retreat in early July has around 70% of chance to belong to the full winter-freezing cluster which shows ice-free conditions in summer. A first date of advance in early September has around 95% of chance to belong to the full winter freezing cluster which presents fully ice covered condition in the following winter.

Another important aspect of our analysis is that a given seasonal cycle can be in between two or more seasonal cycle centroids. We therefore believe that a probabilistic view when dealing with clustering is important. By analysis the evolution of the pan-Arctic clusterings over the 1979-2023 period, we show that the probability to belong to the permanent sea-ice seasonal cycle has decreased by 3.1 %/decade which is compensated with an increase of probability to belong to the open-ocean cluster (1.6 % per decade), the full winter freezing cluster (1.1 % per decade) and to a smaller extent to the partial winter-freezing cluster (0.5 % per decade). Regional shift in the clusters occurs over time. In general, the permanent sea-ice retraction from the Pacific side is compensated by the full winter-freezing cluster and the open-ocean

cluster expansion in the Atlantic side is compensated by loss of the partial winter-freezing cluster.

The added value of our description compared to the MIZ category (sea-ice concentration between 0.15 and 0.8) is the new classes of partial and full winter freezing. This could be important for sea-ice dynamics and forecasting understanding. Our result suggests that the trend is primarily controlled by the trend of the more abrupt melting and growth seasonal cycle (full winter-freezing cluster) compared to the trend of the quasi-sinusoidal sea-ice seasonal cycle (partial winter-freezing cluster) or, in other words, that the trend is more likely due to increase of regions with total ice cover in winter with a short ice-free season (2 months, full winter-freezing cluster) than increase of regions with a partial ice cover in winter with a long ice-free season (4 months, partial winter freezing cluster).

We introduce another diagnostic to quantify the regime stability and transition of the Arctic sea ice. The stable region (having always the same dominant cluster for the whole period 1979-2023) predominantly covers the central part of the Arctic Ocean, including the area around the North Pole, following most of the regions covered by permanent sea-ice cluster, as well as the ocean regions in the open-ocean cluster. Smaller regions present stable conditions: the northern Baffin Bay and southeast of Kara Sea dominated by the full winter-freezing cluster and northern Bering Sea associated with the partial winter-freezing cluster. From the Beaufort to the Kara Seas, the southern parts have stabilized (experiencing a new typical seasonal cycle, corresponding to the full winter-freezing cluster) and the northern part have destabilized (losing their typical permanent sea-ice seasonal cycle).

This regionalization suggests a more latitudinal vision of the region. Also, this study calls for pan-Arctic sea-ice thickness observation in order to better understand sea-ice changes.

# 5. Discussion

The k-means clustering of the sea-ice seasonal cycle we applied to the Arctic shares similarities with the analysis of Wachter et al. (2021) for the Antarctic. The main

differences however reside in our use of Mahalanobis distances, to account for the correlation between the months, and the initialization based on equal separation of quantiles for the centroids, to avoid any random aspect in the clustering algorithm. These two choices enable to constrain the clustering with physical features. Besides, by the use of the Silhouette coefficient, we found the Arctic is best described with a number of clusters of 3 (the open-ocean has been added afterward). This number has also been found by Fuckar et al., (2016) using a suite of indices (Krzanowski-Lai, Calinski-Harabasz, Duda-Hart J index, Ratkowsky-Lance, Ball-Hall, point-biserial, gap statistic, McClain-Rao, tau and scatter-distance index) onto detrended sea-ice thickness of an ocean-sea ice general circulation model. In contrast with Fuckar et al., (2016) that calculated time series of occurrences of clusters based on the resemblance of the pan-Arctic pattern, our probabilistic method defines a time series of probability of occurrence of each cluster at the grid cell scale. This enables us to study the spatial evolution of the cluster areas, and therefore define spatio-temporal regions that share a common feature (in our case sea-ice seasonal cycle).

Our clustering approach is complementary to diagnostics involving the dates of melting and freezing onsets, which have been used to quantify changes in the duration and shift of ice-free seasons at the pan or regional Arctic scales (Markus et al., 2009; Stammerjohn et al., 2012; Parkinson 2014; Johnson & Eicken, 2016; Stroeve et al., 2014; Lebrun et al., 2019). Instead, our method enables us to target regions experiencing a redistribution to another typical seasonal cycle representing longer and ice-free seasons, and retrieve the year of the shift. Another advantage is that we do not use any arbitrary cutoff of SIC. Additionally, our diagnostic delimits regions with the same sea-ice seasonal dynamics. The major drawback of our approach resides in the exact grid point quantification of the real seasonal cycle features (such as ice-free duration), as we gather grid cells within a type represented by a single seasonal cycle (the centroid). However, considering the full seasonal cycle gives useful information, as its derivative gives the period of melting and growth. Therefore, the two diagnostics complement each other nicely.

By doing the diagnostic of the trend in the length of the sea-ice season for the period 1979-2013, Parkinson (2014) shows that the length of the ice season has shortened in almost all the coastal regions (around -10 days/decade with a maximum -30 days/decade in the northern Chukchi Sea and around -50 days/decade in the northern Barents Sea), the main exceptions being the Bering Sea, portions of the Canadian Archipelago (around +10 days/decade) and the central Arctic where the sea-ice season duration remain unchanged over the period. Similar features are obtained in Lebrun et al., (2019) who considered the period up to 2015. Also, Lukovich and Barber (2007) examination of spatial coherence in SIC anomalies indicates that maximum SIC anomalies prevail near the Kara Sea, Beaufort Sea, and Chukchi Sea regions during late summer/early fall from 1979 to 2004. All these studies are consistent with our results showing a decrease in probability for the permanent sea-ice cluster of about 3.1% per decade, especially in coastal regions of the Pacific side of the Arctic, leading to a shortening of the seasonal cycle. Moreover, we were able to show that this regime transition occurs in a smooth northward propagation.

Our clustering approach suggests that the first date of freezing and advance could be key for predicting ice conditions around 6 months in advance. This feature follows a physical behaviour of sea-ice shown by Stammerjohn (2012) and Stroeve et al. (2016). They found strong correlations between the dates of the spring sea-ice retreat and subsequent autumn sea-ice advance (i.e., over the summer), indicating that an early sea-ice retreat is often followed by a late autumn sea-ice advance and conversely, a late sea-ice retreat is often followed by an early autumn sea-ice advance. Indeed, consistent with our clustering analysis, the partial winter-freezing cluster has an early sea-ice retreat (in March) and late autumn sea-ice advance (mid-October) while the full winter-freezing cluster has a late sea-ice retreat (in April) and early autumn sea-ice advance (mid-September). Therefore, this simple model suggests that the first date of retreat could be a good indicator for ice-free conditions the following summer and the first date of advance a good indicator for fully ice cover conditions the following winter. A redefined model which quantifies this without taking into accounts specified clustering is out of the scope of the study. An example of such studies has been done in the Antarctic and shows that at interannual timescales, retreat date anomalies are constrained by seasonal maximum ice thickness (Himminch et al., 2025)

and the advance date is controlled by the timing of sea-ice retreat through heat stored in the summer ocean mixed layer (Himmich et al., 2023). In the Arctic, Gregory et al., 2020 by setting up a complex network statistical approach, shows good predictive skills for regional September SIE from previous June SIC, especially toward the Pacific sector.

Concerning the growth and melting of sea-ice, Parkinson et al., 1999 and Parkinson and Cavalieri, 2008 showed that the seasonal decay of sea ice extent is gradual during early summer and then accelerates during the remaining summer months, whereas wintertime growth is most rapid in early winter. A standard explanation suggests that this asymmetry between seasonal growth and decay is caused by rapid temperature changes driven by air masses from the Eurasian continent (Peixoto and Oort, 1992). Here this asymmetry in the seasonal cycle is seen only for the permanent sea-ice cluster and full winter freezing cluster, suggesting that the partial winter sea-ice is driven by another driver. The full winter-freezing cluster (with no sinusoidal feature) is more likely present along the Arctic coastline than the partial winter-freezing cluster (with a sinusoidal feature). The reason for this spatial repartition could be explained by the fact that the sinusoidal feature of the sea-ice seasonal cycle is linked to the ability of the ice to freeze and expand freely, without being blocked by land, as suggested by Eisenman (2010).

A limitation of the study is the fact that the method accounts solely for the area between the centroid and the seasonal cycles to define the clusters, meaning that there is no constraint to have the same maximum and minimum to belong to one cluster. However, if the shift of minimum or maximum is large, the area will largely increase which prevents having a large discrepancy between the maximum and minimum of the seasonal cycles and their respective centroids. Another limitation of this study is that sea-ice dynamics are analysed using SIC rather than sea-ice volume (which would better represent sea-ice behaviour, including growth and melting), due to the lack of robust and long-term sea-ice thickness data.

The introduction in this paper of the clustering of the Arctic sea-ice seasonal cycle, with its statistical aspect, can provide an approach to validate the dynamics of sea-ice in climate models. Indeed, applying the clustering method described here to

models could inform if a given model has the same number of optimal clusters and the types of seasonal cycles as the one obtained from observations. It could also be used to answer how different clusters will be distributed for different future scenarios. Overall, this methodology is transposable to other variables to better answer its past and future variability in a robust statistical framework. These research findings which are relevant for climate models and process understanding, can also provide useful information for forecast application, guiding ecosystem conservation efforts, and thus related policy-making planning.

## Author's contribution

All authors contributed to the conceptual design of the study and the interpretation of the results. AS, PT, and FS established the methodological framework. AS developed the code, generated the figures, and drafted the initial version of the article. PT, FS, and CL carefully revised the paper contributing to its improvement.

## Financial support

This study is funded by ANR and France 2030 through the project CLIMArcTIC (grant ANR-22-POCE-0005)

## Competing interests

The contact author has declared that none of the authors has any competing interests.

## Code and data availability

The code developed for this study (clustering, diagnostics and plots) is available for download at https://github.com/amelie-simon-pro/SIC_Clustering. We utilized Mistral (https://chat.mistral.ai/chat) and ChatGPT (https://chat.openai.com/) to assist in generating small portions of the code, which we subsequently adapted for our script. The daily SIC satellite data from the National Snow and Ice Data Center (NSIDC) are openly available and can be found at https://doi.org/10.7265/efmz-2t65 (Meier et al., 2021)

## Acknowledgement

We enthusiastically thank the four reviewers for their very constructive comments that helped to improve the paper.

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
