# Peer review of "Arctic regional changes revealed by clustering of sea-ice observations"

_EGUsphere, 2025_

## Referee Comment (RC1)

4 April 2025

Review of "*Arctic regional changes revealed by clustering of sea-ice observations*" by Simon et al.

This review is co-signed by François Massonnet and Noé Pirlet (UCLouvain)

**Summary.**
The authors use an unsupervised machine learning method, namely k-means, to identify regimes of Arctic sea ice concentration (SIC) variability based on the seasonal cycle. They use Mahalanobis distances instead of classical Euclidean distance, to account for the correlation between the months, and the initialization based on equal separation of quantiles for the centroids, to avoid random aspects in the clustering algorithm. They report the mean state and variability / trends of the sea ice state when categorized with this approach.

**Novelty.**
The work provides interesting insights on the dynamics of Arctic sea ice and is a good topic for a journal like The Cryosphere.

**Positioning.** There have been early studies of clustering that are not cited, namely Fuckar et al. (2016) and Lukovich and Barber (2007). The authors should also cite Raphael and Hobbs (2016), since it is an (Antarctic) study that defined regions based on the behavior of sea ice concentration.

**Methodological questions**
Currently our main obstacle to understanding the research is methodological. Our impression is that the manuscript would deserve to have a better description of the details of the methods of clustering, because as such, we would feel unable to reproduce the results due to many missing details. Since we are unsure about several methodological aspects, we do not comment much on the science itself yet – maybe at the next iteration of the review. Below, we identify several places where we think an improvement could be made.

- L. 224: if we understand correctly, there are as many such matrices of size 73 by 1 123 710, as there are years (45), is that correct? So the k-means clustering is applied on all years individually, which allows producing time series. We think it would be good to mention that already here.
- L. 229: "It is an iterative method that minimizes a cost function being the sum of the squared distance between each seasonal cycle and its nearest cluster center (also called centroid)". Here, it would be useful to write ("in a sense to be defined later") after "distance" because it took us some time to understand how a 73-time frame seasonal cycle could be located in the state space. Also, it could be mentioned here that at each iteration, the coordinates of the clusters are updated.
- L. 225-229. We feel that a methodological figure could help here, showing how the SIC fields are arranged in a matrix, how this matrix represents a series of

points in a 73-dimensional space (you can work in 2-D for illustrations), and how the centroids evolve at each iteration, before a remapping is done in physical space. The Cryosphere is a journal where methods from data science can appear new to some readers.

- L. 234-236. Has the method of equal quantile separation been tested elsewhere (if so, please cite a relevant study) or is it something that the authors are proposing in this study?

- L. 234-236. It is not entirely clear how quantiles are calculated on data that has 73 dimensions (and what "quantile separation", line 235, really means)? We understand that it is possible to compute the distance between any two pairs of points by applying the Mahanabis distance on the two 73-long vectors; do the authors then sort all mutual distances and define the quantiles based on that? If so, how do they revert to one centroid given that the distance involves two points? This method is not described in a way that would allow reproducing the results. Please clarify this part.

- L. 246-260. It is not entirely clear how the correlations of Fig. 2 are calculated. If we consider, for example, the entry that connects the first 5 days of February and the first 5 days of June, then what exactly is calculated? Do the authors first average SIC in space and then compute correlation over all pairs of these averages on the 45 years? Do the authors compute correlation over space (in that case, do they stack the 45 years on each other, effectively producing two vectors of 45 times 1 123 710 length? Also isn't there an issue with using correlation here, since most points are either at 0 or at 1 ? A scatter plot would reveal something very different from what a correlation aims to capture from a standard cloud of points. Can the authors elaborate a bit on these two points? Could the authors show the histograms of SIC > 55°N for the 15th of each month, for example?

- Fig. 2: the Figure caption is too short, but instead should allow reproducing the figure unambiguously.

- L. 246-260. These correlations are computed without deseasonalizing nor detrending the data, is that correct? When one applies other data reduction methods like PCA/EOFs, the forced variability is removed first – is it the case here?

- L. 264: what does "normalizing by the correlation matrix" mean here? Divide entry-wise by each element? If multiplying by the inverse of this matrix, then use this phrase – one cannot "divide" by a matrix. Also in L. 256 we read "normalized by the inverse of the correlation matrix" while L. 264 "we read "normalized by the correlation matrix". Could the authors clarify this?

- L. 373: please explain how $M_{cor}$ is constructed (related to our other comment on the corresponding figure).

- L. 379-382: So there the authors used fuzzy k-means but then in the previous section, it was not fuzzy ? This is not very clear, and calls for a better justification of the transition between these two sections. Also, is there an objective criterion to prefer fuzzy k-means over crisp assignments? In general, how do we measure whether the k-means did a good job or not (is it with the Silhouette metric, and if so, what is a "good" clustering)?

**Presentation**

- L. 73 : We think « sea-ice » with a dash should be used when it is used as an adjective; when used as a noun, it should be « sea ice »
- L. 140-157: on the regionalization: it is not entirely clear why this paragraph is here. As we understand, no regionalization is required since the k-means method picks the optimal clusters which can then be used to defined physically-relevant regions. We would propose to move this paragraph to the discussion, since the regionalization is more an outcome of the work than a pre-requisite.
- L. 158-172: the emphasis on regionalization for this paragraph (which follows another paragraph on regionalization) shifts somewhat the research question from the initial goal (understanding the physical regimes) to another goal (defining objective geographic boundaries). We would propose to focus the work on the former, and to discuss later in the text how the k-means can also be seen as a way to provide physically-based regions based on distinct sea ice dynamics. Of course, identifying spatial clusters and delimiting regions are two tasks that go hand in hand, but to us the main research question is not always clear.
- L. 200 to 202: If 5-day mean shows similar results than for 1 day why not keeping 1 day as temporal resolution ? Could the authors precise their reasons for using 5-day mean instead of 1 day?
- Fig. 9 is never cited
- On printed sheets of paper, the figures do not render very well.
- Line 358 to 359. The transition from the section 3.1 to 3.2 is quite enigmatic. Section 3.1 seems to deal with deterministic clustering where each point is assigned to a cluster while section 3.2 deals with probabilistic clustering. We would propose to put the explanation of lines 362-363 at the end of the section 3.1 to make a smoother transition.
- Line 392: Could the authors explain why they chose to divide the period into 3 sub-periods of 15 years ? It seems a bit arbitrary and it's hard to understand where the authors are going with this sub-time division. Could the authors add a bit of context to introduce that choice?
- Line 456 to 459 : We find it unclear whether the probabilities of belonging to a cluster given in this paragraph are the average of the probabilities over all the years or just one year? Looking at Figure 7, it seems that this is for 1979. If that's the case, why take one year rather than the average? Could the authors clarify this?
- Line 511. We are unsure about what the authors are trying to imply/measure by defining "stable", "stabilization", "unstable" or "destabilization" regimes. In particular, they should relate those regimes to something known from the literature or give more interpretation because this notion and its application come somewhat out of the blue.

**Minor comments**

- L. 43 « optimal » should be accompanied by « (in the sense of statistical dissimilarity) » or something like that, because otherwise this word can be wrongly interpreted.

- L. 45. The use of the phrase "ice-free conditions" should be taken with care here and throughout the manuscript, since ice-free has a well-defined meaning in the climate projection literature (namely, sea ice extent < 1 million km$^2$). What exaclty is ice-free in the context of this study? We also propose to add a horizontal line on Fig. 4a to represent the chosen threshold, in order to quickly see when the seasonal cycle shows ice-free conditions.
- l. 43-45: When reading this part of the abstract, it sounds like the authors are re-discovering two regimes that are well-known (open-ocean and permanent ice) and two intermediate regimes where ice is present seasonally. Since the work goes deeper than re-inventing these regimes, we would propose to describe the clusters with more physical interpretation and to highlight what the k-means analyses have allowed to do, that the human eye is not able to see (i.e., what is the added value of an algorithm that can deal with large amounts of data in a high dimension space)
- l. 50 likelihood reduction → likely reduction? Or do the authors mean that the likelihood of this regime in the Canadian side has reduced? Maybe likelihood should be changed by "probability of occurrence" to make things clearer?
- L. 52. You mean that spatial redistributions occur within the four clusters ?
- L. 53-55 : it is grammatically strange to write that a "sea" is stabilizing. It is the state characterized by a given cluster that becomes less frequent, in a given region/sea.
- L. 54-55: be consistent grammatically: « have destabilized » … « have stabilized »…?
- L. 113 : % relative to what ?
- L. 116 : maybe « and do not consider changes in the underlying processes » ?
- L. 125: it would be worth mentioning that these studies have highlighted an asymmetry in the trend of retreat vs advance (see Lebrun et al. In particular). Indeed, the current manuscript also reveals an asymmetry in the seasonal cycle, so there is a nice connection to be made here.
- L. 128 it is true that these studies do not inform on the sea ice dynamics including melt and growth behaviours, but does the present study do so, given that it does not seek to study the mass balance terms nor the time derivatives of sea ice concentration?
- L. 131-132. The statement that previous studies have not directly considered the full sea ice seasonal cycle, is not entirely correct. For example, previous studies looking at SIE report the seasonality of trends, of the mean state, of the variability. But they consider each point of the season separately, while in the present study the seasonality is accounted implicitly for while producing the clusters, through the correlation matrix.
- L. 216 a "non-zero" seasonal cycle could be interpreted differently by different readers. We assume you mean "having at least a non-zero value for SIC throughout the year" ?
- Line 298: "one of the two outputs" leads the reader to wonder what the other output might be, and it is not displayed in the same paragraph. It is only introduced at line 347. Please mention that the other output (the connection of each grid point to a cluster) is studied later.
- L. 302: follows → lags?

- L. 339 : apparition → appearance ?
- Fig. 4 : the caption is not sufficient to understand what is plotted. Panel (a) shows the average concentration over each cluster, is that correct? How is the time dimension dealt with here? Are the average SIC determined for each cluster, then averaged in time?
- Fig. 4b. have the authors plotted these maps for iconic years like 2007 (big ice arch between the Greenland and the Kara Sea or 2012 (absolute minimum)? That would be interesting to see to what extent the method captures the physics of those events.
- Line 370 centroids → centroid
- Fig. 5: From a quick look, one could argue that the method is not really able to sort the data, as all four clusters have ~25% probability near the ice edge. We were both surprised to see so little variability in the maps, as they seem to have quite homogenous spatial distributions of probabilities.
- Fig. 7. How is the "total probability" calculated here? Each spatial point has been assigned a probability, are these probabilities then averaged in space? Does it make physical sense to average probabilities in space?
- Fig. 8: y-label units should be $m^2$ not $km^2$
- Fig. 8: please repeat in the caption how Open-ocean, MIZ and Packed ice are defined.
- Fig. 8: how do we explain that there is much less interannual variability in the (b) panel (that is, hand-made clustering) compared to (a) (k-means)?
- Lines 505-509: it would be good to interpret the fact that 2007 is only visible for the permanent cluster, in terms of existing literature that studied this event. In fact, we would argue that the signature is also visible in the full winter-freezing cluster, in "negative". That is, the 2007 event seem to be interpretable as a strong drop of perennial ice but also as a strong surge of the full winter freezing cluster. Why is it so? We would have expected to see a surge of the open water cluster in 2007, not the winter cluster.

Fučkar, N. S., Guemas, V., Johnson, N. C., Massonnet, F., & Doblas-Reyes, F. J. (2016). Clusters of interannual sea ice variability in the northern hemisphere. *Climate Dynamics*, *47*(5), 1527–1543. https://doi.org/10.1007/s00382-015-2917-2

Lukovich, J. V., & Barber, D. G. (2007). On the spatiotemporal behavior of sea ice concentration anomalies in the Northern Hemisphere. *Journal of Geophysical Research: Atmospheres*, *112*(D13). https://doi.org/10.1029/2006JD007836

Raphael, M. N., & Hobbs, W. (2014). The influence of the large-scale atmospheric circulation on Antarctic sea ice during ice advance and retreat seasons: RAPHAEL AND HOBBS; ANTARCTIC SEA ICE ADVANCE AND RETREAT. *Geophysical Research Letters*, *41*(14), 5037–5045. https://doi.org/10.1002/2014GL060365

---

## Referee Comment (RC2)

Review
EGUsphere
Manuscript Number: egusphere-2025-704
Title: Arctic regional changes revealed by clustering of sea-ice observations
Authors: Amélie Simon, Pierre Tandeo, Florian Sévellec, Camille Lique

*General comments:*

Arctic sea ice is one of the most affected components by climate change. Understanding its evolution over the last 40 years is key to prepare for further amplification of Arctic warming. As such, physical variable can be used to examine the different behaviors of sea ice. The present study addresses the following question: What insights can be gained about sea ice behavior trends through the application of clustering techniques?

In this work, the authors relied on sea ice concentration (SIC) seasonal cycle to identify 4 sea ice behaviors instead of the classical approach of splitting the Arctic based on geographical regions. The study spams from 1979 to 2023, with daily values averaged as 5-day mean and SIC are from passive microwave satellite observations. The four optimal clusters can change over time and are identified as: open ocean, permanent sea-ice, partial winter freezing and full winter freezing. The authors show the long term changes of the 4 types of seasonal cycle of SIC. They also introduce transitions from one behavior to another, which are either stabilization (typically any ice regime to open ocean) and destabilization (typically permanent ice to any winter freezing regime).

The paper is well structured and clear, which makes it pleasing to read. The context and method are thoughtfully described. The results are clearly explained, properly analyzed. Especially the section 3.4.2. which shows a very interesting analysis.
In my opinion, this is a great paper which could be improved by discussing the limitations of this study in section 4. Some additions can be added to the text for clarification.

*In the following pages, I address several points that requires the authors' attention and I hope they will help improving the present manuscript:*

The SIC product is presented, yet no limitations nor assumptions made to obtain SIC are presented. Is there any reason to pick this product compared to another SIC product? I believe 1-2 sentences of this topic would bring perspective to the text and remind the reader that this dataset differs from reality. Either in section 2.1 or section 4. Is the product consistent over the 40 years? What is the uncertainty of the measure?

At several occasions (l.216, l.378, l.422, l.600, l.621, l.658), the authors write 'sea-ice seasonal cycle'. In my understanding, the seasonal cycle can refer to different variables such as concentration, thickness, albedo, etc. For sake of clarity, I suggest either:
- add a sentence stating that throughout the manuscript, sea-ice will always relate to concentration,
- add 'concentration' to each instance of term 'seasonal cycle'.

The authors uses the clustering analysis to identify sea-ice precursors for one given point (3.1, L. 325-328). While I find this analysis interesting, I believe using the term 'predictor' is misleading. To my understanding, there is no prediction in this manuscript; the dataset is entirely based on past data. Therefore, the clustering outputs do not predict the behavior of sea-ice, but indicate general behavior for a given grid cell. I do agree that the clustering relies heavily on the start of melting and freezing periods. Using the term 'indicator' would remove this potential confusing.

Figure 10 appears before Figure 9 in the text. Figure 9 is actually not cited in the text directly. I would recommend swapping order of Figure 10 and 9, and adding a citation of former Figure 9 in the paragraph between l. 554-562, where the "star" and "triangle" examples are described.

Overall, the section "conclusion and discussion" present the final results and compare them with the literature. This comparison is satisfactory, however it lacks discussion on the limitation of this study regarding the data and the method.

*Need for clarifications:*

On l.128, the authors state that 'usual descriptors… do not account/consider for the full seasonal cycle' of sea ice. I find this statement unclear (also in the abstract., l.40). Are the cited studies only considering part of seasonal cycle (by choice, lake of data, lack of mean) ? Are they using SIC as well or another variable which could be partially unavailable (such as thickness during melting period) ? Why SIE or type of sea-ice can not consider the full sea-ice seasonal cycle (also L.131)?

L. 116: 'do not consider changes in sea-ice features'. "Features" is used several times throughout the manuscript (l. 637), and I think concrete examples of features should be given by the authors at the first reference. By the end of the manuscript, I understand that such 'feature' means the duration of ice season, or duration of open ocean. At my first reading, it could also have been the sea ice thickness distribution or other properties of sea ice which are not tackled here.

L. 317: When writing the dates of melting and freeze up, the authors mention first the permanent, followed by full winter-freezing and partial winter-freezing. However, throughout the paragraph, the seasonal cycles are presented in a different order: open-ocean, partial winter-freezing, full winter-freezing, and permanent ice clusters.
For consistency, I recommend keeping the same order as initially mentioned.

L. 635: "The major limit of our approach": I find that this major limitation is explained relatively shortly. In addition, I do not really understand this sentence. Some rephrasing or additional explanation are necessary. Do you mean to say that a lot of different seasonal cycle of SIC (grid cells) are reduced to one single seasonal cycle through clustering?

*Specific comments:*

L. 136: citation from Lebrun et al. (2019) is missing.

L. 173: "for the first time". Is this really the first time? This is quite surprising! The authors did a great job in showing the potential of such a method.

Few sentences are very long: I could suggest rephrasing them: L. 177, l. 301.

L. 234: 'influencing'. As influence could be positive, I had to keep reading that this influence was undesired, regardless of it being beneficial or not. Maybe use 'biasing'.

L. 238-239: Quantiles are given both with % (33%, 66%) and as float (0.25, 0.50). Please, pick one way and stick with it. I would advise for %, as it is used again later in the manuscript.

L. 248: At the end of "Mahalanobis distance to constrain the clustering with physical information.", I was expecting to find the definition of Mahalanobis distance which is L.255. Please, consider putting the definition as soon as possible.

L. 249: Parenthesis is not closed. "(as shown in Figure 2 by…"

L. 301: "They exhibit the expected physical behavior that, ...maximum solar insolation (Parkinson et al. 1987)." Although I see what the authors want to express, this sentence appears difficult to read and cumbersome. Please, rephrase and make this sentence lighter.

L. 423: "(from no ice to 70% SIC for the partial winter freezing clusters and to 100% SIC for the full winter freezing cluster)". I find this information relevant and I think it would fit better in an addition sentence than in parenthesis.

L. 476: "and to a smaller extent, of the full winter-freezing cluster." What about the decrease in partial winter freezing? It could also be compensated by a gain in full winter-freezing?

Figure 8: There are common markers between the subplots. While the open-ocean category can share the same marker and color, the square and diamond markers indicate different categories in both subplots. I would recommend changing the markers in subplot b) to eliminate any possible confusion (especially if printed in black and white). Additionally, adding a vertical line at the year 2000 can enhance the graphical readability.

L. 498: ") The area of the MIZ" > "t"

L. 508: It would be interesting to add one sentence about the supposed reason why the sea-ice loss signature is only visible in the permanent sea-ice cluster. I expected to find this in the discussion section but did not see it.

L. 547: "As shown Figure 10" > "in" missing

L. 612: "the area between the Central Arctic and the open-ocean does not" > I suggest adding "permanent ice" for clarity. " the area between the permanent ice Central Arctic and the open-ocean does not"

L.652: "The year of loss in the likelihood to belong to the permanent sea-ice shows" is heavy to read for my tastes.

---

## Author Comment (AC1)

* * *
Reviewer 2

General comments:

Arctic sea ice is one of the most affected components by climate change. Understanding its evolution over the last 40 years is key to prepare for further amplification of Arctic warming. As such, physical variable can be used to examine the different behaviors of sea ice. The present study addresses the following question:

What insights can be gained about sea ice behavior trends through the application of clustering techniques? In this work, the authors relied on sea ice concentration (SIC) seasonal cycle to identify 4 sea ice behaviors instead of the classical approach of splitting the Arctic based on geographical regions. The study spams from 1979 to 2023, with daily values averaged as 5-day mean and SIC are from passive microwave satellite observations. The four optimal clusters can change over time and are identified as: open ocean, permanent sea-ice, partial winter freezing and full winter freezing. The authors show the long term changes of the 4 types of seasonal cycle of SIC. They also introduce transitions from one behavior to another, which are either stabilization (typically any ice regime to open ocean) and destabilization (typically permanent ice to any winter freezing regime).

The paper is well structured and clear, which makes it pleasing to read. The context and method are thoughtfully described. The results are clearly explained, properly analyzed. Especially the section 3.4.2. which shows a very interesting analysis.

In my opinion, this is a great paper which could be improved by discussing the limitations of this study in section 4. Some additions can be added to the text for clarification.

Thank you very much. We have added some text in several sections for clarification, in particular for discussing the limitations of our study.

In the following pages, I address several points that requires the authors' attention and I hope they will help improving the present manuscript:
The SIC product is presented, yet no limitations nor assumptions made to obtain SIC are presented. Is there any reason to pick this product compared to another SIC product? I believe 1-2 sentences of this topic would bring perspective to the text and remind the reader that this dataset differs from reality. Either in section 2.1 or section 4. Is the product consistent over the 40 years? What is the uncertainty of the measure?

Thank you. To make the most of our data driven model, we need the largest amount of data. To the best of our knowledge, there are two satellite data products having daily SIC starting in 1979: NSIDC and OSI SAF (EUMETSAT). We used NSIDC as it is commonly used for climate studies while OSI SAF is commonly used for operational studies.

We now say in section 2.1:" Measurement uncertainties are highest at low SIC, where satellite signals are often influenced more by atmospheric and surface conditions—such as clouds, water vapor, melt on the ice surface, and changes in the character of the snow and ice surface—than by the actual presence of ice. "

At several occasions (l.216, l.378, l.422, l.600, l.621, l.658), the authors write 'sea-ice seasonal cycle'. In my understanding, the seasonal cycle can refer to different variables such as concentration, thickness, albedo, etc. For sake of clarity, I suggest either:
- add a sentence stating that throughout the manuscript, sea-ice will always relate to concentration,
- add 'concentration' to each instance of term 'seasonal cycle'.

Thanks. We now say: "In this paper, we determine Arctic regions based on statistically different sea-ice concentration seasonal cycles, and describe Arctic changes through the time evolving borders. We identify for the first time spatio-temporal regions of the Arctic based on the variability of the seasonal cycle of Arctic sea-ice concentration"

And in the methodology section 2.1, we now say: "Throughout the manuscript, sea-ice will always relate to concentration."

The authors uses the clustering analysis to identify sea-ice precursors for one given point (3.1, L. 325-328). While I find this analysis interesting, I believe using the term 'predictor' is misleading. To my understanding, there is no prediction in this manuscript; the dataset is entirely based on past data. Therefore, the clustering outputs do not predict the behavior of sea-ice, but indicate general behavior for a given grid cell. I do agree that the clustering relies heavily on the start of melting and freezing periods. Using the term 'indicator' would remove this potential confusing.

Thank you very much. We agree. We have changed the term predictor to indicator. Also we substantially pushed further the analysis.

We now say:

"... Therefore, it seems that, for ice-free conditions in summer, the first date of freezing is a good predictor for the appearance of full ice conditions in the next winter.

However, this suggestion relies solely on the shape of the four types of seasonal cycles but to properly quantify this, the spread must be taken into account. Figure S2 displays the spread of the seasonal cycle by plotting the quantiles 0.1, 0.5 and 0.9 of each cluster. To verify our hypothesis on sea-ice predictors, we account for the spread of the date of retreat and date of advance for each cluster. To do so, we calculate the first date of retreat (the first date after the maximum SIC that is below 0.9) for each seasonal cycle experiencing fully ice covered conditions (having at least one value above 0.99 during the year). We also calculate the first date of advance (the first date after the minimum SIC that is above 0.1) for each seasonal cycle experiencing ice-free conditions (having at least one value below 0.01 during the year). For these calculations, seasonal cycles have been temporally filtered using a 15 days sliding window in order to get rid of short-term dynamical ice events, as done in Lebrun et al., (2019). To circumvent the effect of the discontinuity between 31 December and 1 January, we define the origin of time in May for the calculation of the date of advance. We then label each first date of retreat and first date of advance for each seasonal cycle with its corresponding cluster according to our clustering analysis (Figure 4a).

The normalized probability over each cluster of the first date of retreat and first date of advance at each date is shown Figure 5. This figure also displays the total number of the first date of retreat and the first date of advance of all clusters for each date. If the first date of retreat occurs between January and April, there is around 95% of chance to belong to either the open-ocean cluster, the partial winter-freezing cluster or full winter freezing cluster, which all present ice-free duration in the following summer. However, this situation did not often occur, as the total first date of retreat happening in this period is unlikely (solely 5% of first date of retreat for all clusters). The first date of retreat is more likely to occur between the beginning of April and August, as within this period around 90% of the total date of retreat for all clusters exist. A first date of retreats in early June has solely around 10% of chance to belong to the permanent sea-ice cluster which do not present ice-free conditions in summer while a first date of retreat in early July has around 70% of chance to belong to the full winter-freezing cluster which shows ice-free conditions in summer.

The first date of advance is more likely to occur between the beginning of August until the beginning of January, as within this period around 90% of the total date of advance for all clusters exist. A first date of advance in early September has around 95% of chance to belong to the full winter freezing cluster which present fully ice covered condition in the following winter, while a first date of advance in early

 November has around 80% of chance to belong to the partial winter freezing or open
ocean clusters which do not show fully ice covered conditions in the following winter.
Therefore, this simple model suggests that the first date of retreat could be a good
predictor for ice-free conditions the following summer and the first date of advance a
good predictor for fully ice cover conditions the following winter. "

[Figure]

Figure 5: Normalized probability of the first date of retreat (panel a) and first
date of advance (panel b) for each cluster. The solid lines with star markers are the
total number of first dates of retreat and first dates of advance for each date. The

 green circle markers (start date) and green square markers (end date) cover the shortest period where around 90% of the first date of retreat, respectively the first date of advance,  for all clusters occurs."

Figure 10 appears before Figure 9 in the text. Figure 9 is actually not cited in the text
directly. I would
recommend swapping order of Figure 10 and 9, and adding a citation of former
Figure 9 in the paragraph between l. 554-562, where the "star" and "triangle"
examples are described.
We have reorganized the figure number and now say:"Figure 10 illustrates how we
define the stabilization and destabilization labels."
Overall, the section "conclusion and discussion" present the final results and compare
them with the literature. This comparison is satisfactory, however it lacks discussion
on the limitation of this study regarding the data and the method.
Thank you very much. We have added a new paragraph in the discussion . We now
say: "A limitation of the study is the fact that the method accounts solely for the area
between the centroid and the seasonal cycles to define the clusters, meaning that
there is no constraint to have the same maximum and minimum to belong to one
cluster. However, if the shift of minimum or maximum is large, the area will largely
increase which prevents having a large discrepancy between the maximum and
minimum of the seasonal cycles and their respective centroids.  Another limitation of
this study is that sea-ice dynamics are analysed using SIC rather than sea-ice volume
(which would better represent sea-ice behaviour, including growth and melting), due
to the lack of robust and long-term sea-ice thickness data. "
Need for clarifications:
On l.128, the authors state that 'usual descriptors... do not account/consider for the
full seasonal cycle' of sea ice. I find this statement unclear (also in the abstract., l.40).
Are the cited studies only considering part of seasonal cycle (by choice, lake of data,
lack of mean) ? Are they using SIC as well or another variable which could be partially
unavailable (such as thickness during melting period) ? Why SIE or type of sea-ice can
not consider the full sea-ice seasonal cycle (also L.131)?
These studies do not directly consider the full sea-ice concentration seasonal cycle, in
comparison to our study that directly uses the shape of the seasonal cycle in the
calculation to analyze Arctic changes. However, these other methods highlighted
changes in the shape of the sea-ice seasonal cycle, in an indirect way.
We say: "These three ways of describing the variations in Arctic SIC (trend of

SIE,  sea-ice age, ice-free duration), without considering directly the full sea-ice seasonal cycle, have nonetheless highlighted changes in the shape of the sea-ice seasonal cycle: (i) the trend in SIE depends on the season, being maximum in late summer (Fox-Kemper et al., 2021 in IPCC, their Figure 9.13; Meier and Stroeve, 2022), (ii) Arctic sea ice has shifted to younger ice between 1979 and 2018 (IPCC, 2019) and (iii) the trend of later ice advance is expected to eventually double that of earlier retreat over this century, shifting the ice-free season into autumn (Lebrun et al., 2019). Here, in this paper, we describe the evolution of the Arctic by delimiting spatio-temporal regions having a common type of seasonal cycle. "

L. 116: 'do not consider changes in sea-ice features'. "Features" is used several times throughout the manuscript (l. 637), and I think concrete examples of features should be given by the authors at the first reference. By the end of the manuscript, I understand that such 'feature' means the duration of ice season, or duration of open ocean. At my first reading, it could also have been the sea ice thickness distribution or other properties of sea ice which are not tackled here.

We now say:"The major drawback of our approach resides in the exact grid point quantification of the real seasonal cycle features (such as ice-free duration)".

L. 317: When writing the dates of melting and freeze up, the authors mention first the permanent, followed by full winter-freezing and partial winter-freezing. However, throughout the paragraph, the seasonal cycles are presented in a different order: open-ocean, partial winter-freezing, full winter-freezing, and permanent ice clusters.
For consistency, I recommend keeping the same order as initially mentioned.

In the new version, we have removed this paragraph. We now say:"Considering a given location fully ice-covered in a given winter, our clustering results suggest that when the sea ice starts to melt in April, the seasonal cycle belongs to the full winter-freezing cluster and be ice-free the next summer. In contrast, when the melting starts one month later (in May) the seasonal cycle belongs to the permanent sea-ice cluster and the considered location will not be ice-free in summer. Besides, the freezing date for areas free of ice could differentiate between the partial winter-freezing and full winter freezing clusters and subsequently predict full ice conditions in the following winter. In our clustering, a freezing starting in October totally freezes in winter which is not the case if the freezing starts in November, having a maximum of about 70% SIC in March. Therefore, it seems that, for ice-free conditions in summer, the first date of advance is a good indicator for full ice conditions in the next winter. "

L. 635: "The major limit of our approach": I find that this major limitation is explained relatively shortly. In addition, I do not really understand this sentence. Some rephrasing or additional explanation are necessary. Do you mean to say that a lot of different seasonal cycle of SIC (grid cells) are reduced to one single
seasonal cycle through clustering?
Yes exactly. We now employ "drawback" instead of "limit" and wrote one specific
paragraph for the limitation of our study.
We now say:"A limitation of the study is the fact that the method accounts solely for
the area between the centroid and the seasonal cycles to define the clusters, meaning
that there is no constraint to have the same maximum and minimum to belong to one
cluster. However, if the shift of minimum or maximum is large, the area will largely
increase which prevents having a large discrepancy between the maximum and
minimum of the seasonal cycles and their respective centroids.  Another limitation of
this study is that sea-ice dynamics are analysed using SIC rather than sea-ice volume
(which would better represent sea-ice behaviour, including growth and melting), due
to the lack of robust and long-term sea-ice thickness data. "
Specific comments:
L. 136: citation from Lebrun et al. (2019) is missing.
Done.
L. 173: "for the first time". Is this really the first time? This is quite surprising! The
authors did a great job in showing the potential of such a method.
We do not say that it is the first time that the method is used.
We say:"In this paper, we identify for the first time spatio-temporal regions of the
Arctic based on the natural variability of the seasonal cycle of Arctic sea-ice."
Few sentences are very long: I could suggest rephrasing them: L. 177, l. 301.
We have shortened L177 and now say:"In section 3, we first analyze the clustering
outputs of the Arctic sea-ice seasonal cycle (3.1), then examine the probability to
belong to each cluster (3.2), and finally investigate the regime stability and transition
(3.3)."
For L 301, we say:"They exhibit the expected physical behavior that, due to the
thermal inertia of the ice and indirect processes involving the ocean and atmosphere,
the maximum sea-ice coverage (in March) follows the minimum solar insolation by a
lag of around 3 months, and the minimum sea-ice coverage (in September) occurs
around 3 months after the maximum solar insolation (Parkinson et al. 1987). "
We agree that the sentence is long but we think it is clear enough.
L. 234: 'influencing'. As influence could be positive, I had to keep reading that this
influence was undesired, regardless of it being beneficial or not. Maybe use 'biasing'.
We choose impacting as different realization gives different results. We now say:
"The initialization of centroids coordinates using k-means++ concept (the first centroid is chosen randomly, the second is the farthest-away, the third the farthest-away of the first and second, and so on) has been tested and is partly impacting our results."

L. 238-239: Quantiles are given both with % (33%, 66%) and as float (0.25, 0.50). Please, pick one way and stick with it. I would advise for %, as it is used again later in the manuscript.

Thank you. We now use the "float" way everywhere in the manuscript.

L. 248: At the end of "Mahalanobis distance to constrain the clustering with physical information.", I was expecting to find the definition of Mahalanobis distance which is L.255. Please, consider putting the definition as soon as possible.

Thank you. We now say straight forward that it uses the correlation matrix:" we choose to use the Mahalanobis distance (using the correlation matrix) to constrain the clustering with physical information."

L. 249: Parenthesis is not closed. "(as shown in Figure 2 by…"

Thank you. Done.

L. 301: "They exhibit the expected physical behavior that, …maximum solar insolation (Parkinson et al. 1987)." Although I see what the authors want to express, this sentence appears difficult to read and cumbersome. Please, rephrase and make this sentence lighter.

We say:"They exhibit the expected physical behavior that, due to the thermal inertia of the ice and indirect processes involving the ocean and atmosphere, the maximum sea-ice coverage (in March) follows the minimum solar insolation by a lag of around 3 months, and the minimum sea-ice coverage (in September) occurs around 3

months after the maximum solar insolation (Parkinson et al. 1987). "

We agree that the sentence is long but we think it is clear enough.

L. 423: "(from no ice to 70% SIC for the partial winter freezing clusters and to 100% SIC for the full winter freezing cluster)". I find this information relevant and I think it would fit better in an addition sentence than in parenthesis.

We think parenthesis are a good choice to keep the sentence concise.

L. 476: "and to a smaller extent, of the full winter-freezing cluster." What about the decrease in partial winter freezing? It could also be compensated by a gain in full winter-freezing?

Thank you. We now say:"The pan-Arctic probability to belong to the permanent sea-ice seasonal cycle has decreased by 3.1 %/decade which is compensated with an increase of probability to belong to the open-ocean cluster (1.6 % per decade), the full winter freezing cluster (1.1 % per decade) and the partial winter-freezing cluster (0.5 % per decade). "

Figure 8: There are common markers between the subplots. While the open-ocean category can share the same marker and color, the square and diamond markers indicate different categories in both subplots. I would recommend changing the markers in subplot b) to eliminate any possible confusion (especially if printed in black and white). Additionally, adding a vertical line at the year 2000 can enhance the graphical readability.

We think we can easily refer to the legend for the markers of each subplot to not be confused. Also we have vertical lines every 5 years, which leads to a vertical line for the year 1999 and 2003. We think adding a line for 2000 will overload the graphic.

L. 498: ") The area of the MIZ" > "t"

Done.

L. 508: It would be interesting to add one sentence about the supposed reason why the sea-ice loss signature is only visible in the permanent sea-ice cluster. I expected to find this in the discussion section but did not see it.

We now use this paragraph to compare our clustering to the MIZ categorization. We now say:"Also, looking at the years with marked extremes in September sea ice extent, (2007, 2012, 2016 and 2020; see introduction), the MIZ categorization shows a transfer of area between the packed ice and the MIZ. In our clustering vision, 2007, 2012 and 2020 show a transfer of area between the permanent sea-ice cluster and full winter-freezing cluster while 2016 show a transfer of area between the full winter-freezing and the partial winter freezing, reflecting different dynamical changes in the sea-ice seasonal cycles. Therefore, our clustering analysis presents a more detailed description of the MIZ category. "

We think further interpretation on why the signature  is only seen in the permanent sea-ice cluster is out of the scope of this work.

L. 547: "As shown Figure 10" > "in" missing

We now say :"As shown in Figure..."

L. 612: "the area between the Central Arctic and the open-ocean does not" > I suggest adding "permanent ice" for clarity. " the area between the permanent ice Central Arctic and the open-ocean does not"

We have removed this sentence in the new version.

L.652: "The year of loss in the likelihood to belong to the permanent sea-ice shows" is heavy to read for my tastes

We now say:"All these studies are consistent with our results showing a decrease in probability for the permanent sea-ice cluster of about 3.1% per decade, especially in coastal regions of the Pacific side of the Arctic, leading to a shortening of the seasonal cycle. "

---

## Author Comment (AC2)

**Answers to review**

**Arctic regional changes revealed by clustering of sea-ice observations**

Amélie Simon[1][2], Pierre Tandeo[1][3], Florian Sévellec[2][3], Camille Lique[2]

[1] IMT Atlantique, Lab-STICC, UMR CNRS 6285, 29238, Brest, France
[2] Univ Brest CNRS Ifremer IRD, Laboratoire d'Océanographie Physique et Spatiale (LOPS), Brest, France
[3] ODYSSEY Team-Project, INRIA CNRS, Brest, France

**Reviewer 1**

This review is co-signed by François Massonnet and Noé Pirlet (UCLouvain)

**Summary.**

The authors use an unsupervised machine learning method, namely k-means, to identify regimes of Arctic sea ice concentration (SIC) variability based on the seasonal cycle. They use Mahalanobis distances instead of classical Euclidean distance, to account for the correlation between the months, and the initialization based on equal separation of quantiles for the centroids, to avoid random aspects in the clustering algorithm. They report the mean state and variability / trends of the sea ice state when categorized with this approach.

**Novelty.**
The work provides interesting insights on the dynamics of Arctic sea ice and is a good topic for a journal like The Cryosphere.

**Positioning.**
There have been early studies of clustering that are not cited, namely Fuckar et al. (2016) and Lukovich and Barber (2007). The authors should also cite Raphael and Hobbs (2016), since it is an (Antarctic) study that defined regions based on the behavior of sea ice concentration.

Thank you very much for these very interesting and relevant articles.

We have now added in the introduction:
"Using an ocean-sea ice general circulation model, Fuckar et al. (2016) performed a k-means cluster analysis on pan-Arctic detrended sea-ice thickness and found that the associated binary time series of cluster occurrences exhibit predominant interannual persistence with mean timescale of about 2 years."

"A statistical regionalization method based on observed SIC has been proposed for Antarctica. Raphael and Hobbs, (2014) isolates regions around Antarctica by using sea ice extent decorrelation length scale and variance. The resulting five sectors exhibit distinct times of sea-ice advance and retreat. Their methodology does not account for the temporal evolution of the sectors."

And in the discussion:
"Besides, by the use of the Silhouette coefficient, we found the Arctic is best described with a number of clusters of 3 (the open-ocean has been added afterward). This number has also been found by Fuckar et al., (2016) using a suite of indices (Krzanowski-Lai, Calinski-Harabasz, Duda-Hart J index, Ratkowsky-Lance, Ball-Hall, point-biserial, gap statistic, McClain-Rao, tau and scatter-distance index) onto detrended sea-ice thickness of an ocean-sea ice general circulation model."

"In contrast with Fuckar et al., (2016) that calculated time series of occurrences of clusters based on the resemblance of the pan-Arctic pattern, our probabilistic method defines a time series of probability of occurrence of each cluster at the grid cell scale. This enables us to study the spatial evolution of the cluster areas, and therefore define spatio-temporal regions that share a common feature (in our case sea-ice seasonal cycle)."

"Also, Lukovich and Barber (2007) examination of spatial coherence in SIC anomalies indicates that maximum SIC anomalies prevail near the Kara Sea, Beaufort Sea, and Chukchi Sea regions during late summer/early fall from 1979 to 2004. All these studies are consistent with our results showing a decrease in probability for the permanent sea-ice cluster of about 3.1% per decade, ..."

**Methodological questions**

Currently our main obstacle to understanding the research is methodological. Our impression is that the manuscript would deserve to have a better description of the details of the methods of clustering, because as such, we would feel unable to reproduce the results due to many missing details. Since we are unsure about several methodological aspects, we do not comment much on the science itself yet – maybe at the next iteration of the review. Below, we identify several places where we think an improvement could be made.

• L. 224: if we understand correctly, there are as many such matrices of size 73 by 1 123 710, as there are years (45), is that correct? So the k-means clustering is applied on all years individually, which allows producing time series. We think it would be good to mention that already here.

The 123710 includes all the seasonal cycles for all grid cells and all years within the same matrix. We reformulate to be clearer: "The input data of our clustering are all the seasonal cycles including every considered grid cell and every year."

To help the reader, we have also produced a schematic of the input data for the clustering Figure 2a.

[Figure]

Figure 2: Schematic of the matrix input data for the k-means clustering (panel a) and correlation matrix of the 5-day mean sea-ice concentration for all non-zero sea-ice seasonal cycle above 55°N (panel b)

• L. 229: "It is an iterative method that minimizes a cost function being the sum of the squared distance between each seasonal cycle and its nearest cluster center (also called centroid)". Here, it would be useful to write ("in a sense to be defined later") after "distance" because it took us some time to understand how a 73-time frame seasonal cycle could be located in the state space. Also, it could be mentioned here that at each iteration, the coordinates of the clusters are updated.

We now say: "It is an iterative method that minimizes a cost function being the sum of the squared distance (distance in a sense that would be defined later) between each seasonal cycle and its nearest cluster center (also called centroid). At each iteration, the coordinates of the centroids are updated.

• L. 225-229. We feel that a methodological figure could help here, showing how the SIC fields are arranged in a matrix, how this matrix represents a series of points in a 73-dimensional space (you can work in 2-D for illustrations), and how the centroids evolve at each iteration, before a remapping is done in physical space. The Cryosphere is a journal where methods from data science can appear new to some readers.

To help the reader, we  have also produced a schematic of the input data for the clustering Figure 2, panel a.

We also now provide the first centroids for the clustering involving 2 to 4 clusters Figure 3, panel b.

[Figure]

[Figure]

Figure 3: Boxplot of the Silhouette coefficient for a number of clusters from 2 to 20. The box extends from the first quartile (0.25) to the third quartile (0.75) of the

Silhouette coefficient. The whiskers indicate the 1st and 99th percentiles. The green-dashed and orange-solid lines indicate the mean and median values, respectively (panel a). Equal quantile separation initialization: centroids of the first iteration of the clustering for a number of cluster of 2, 3 and 4 (panel b)

• L. 234-236. Has the method of equal quantile separation been tested elsewhere (if so, please cite a relevant study) or is it something that the authors are proposing in this study?

We found a reference on that matter (see e.g. Jambudi and Gandhi, 2022). We now say: "The strategy of initialization based on quantiles has been investigated for synthetic and real dataset and has shown a faster convergence compared to Random and Kmeans++ initialization techniques (Jambudi and Gandhi, 2022). "

Jambudi T, Gandhi S (2022) An Effective Initialization Method Based on Quartiles for the K-means Algorithm. Indian Journal of Science and Technology 15(35): 1712-1721.
https://doi.org/10.17485/IJST/v15i35.714

• L. 234-236. It is not entirely clear how quantiles are calculated on data that has 73 dimensions (and what "quantile separation", line 235, really means)? We understand that it is possible to compute the distance between any two pairs of points by applying the Mahanabis distance on the two 73-long vectors; do the authors then sort all mutual distances and define the quantiles based on that? If so, how do they revert to one centroid given that the distance involves two points? This method is not described in a way that would allow reproducing the results. Please clarify this part.

The quantile is solely for the initialization of the centroid's coordinates and is classically computed without using Mahalanobis distance. Instead of initializing randomly the first coordinates of the centroids, we fix it to be the quantiles. The next iteration does not account for quantiles.

We have added Figure 3, panel b to make it more clear.

• L. 246-260. It is not entirely clear how the correlations of Fig. 2 are calculated. If we consider, for example, the entry that connects the first 5 days of February and the first 5 days of June, then what exactly is calculated? Do the authors first average SIC in space and then compute correlation over all pairs of these averages on the 45 years? Do the authors compute correlation over space (in that case, do they stack the 45 years on each other, effectively producing two vectors of 45 times 1 123 710 length? Also isn't there an issue with using correlation here, since most points are either at 0 or at 1 ? A scatter plot would reveal something very different from what a correlation aims to capture from a standard cloud ofpoints.
Can the authors elaborate a bit on these two points? Could the authors
show the histograms of SIC > 55°N for the 15th of each month, for example?
Thank you. We correlate the matrix of Figure 2, meaning we do not average anything,
we directly correlate all non-zero seasonal cycles in time and space.
We have now more explicitly explained the calculation: "The correlation matrix is
computed for all nonzero seasonal cycles for the period 1979-2023 above 55 °N. It is
calculated from the matrix of shape (73, 1123710), having 1123710 value of SIC for
73 dates."
As suggested, we have plotted the histogram of SIC for the 15th of each month that
even if it shows a lot of 0 and 1, shows some nuances. Even if not ideal, we think that
the correlation is acceptable.

[Figure]

• Fig. 2: the Figure caption is too short, but instead should allow reproducing the
figure unambiguously.
We now say in the caption of Figure 2: "Correlation matrix of the 5-day mean sea-ice
concentration for all non-zero sea-ice seasonal cycle above 55°N"
• L. 246-260. These correlations are computed without deseasonalizing nor detrending the data, is that correct? When one applies other data reduction
methods like PCA/EOFs, the forced variability is removed first – is it the case
Here?
The concept is different from PCA/EOF technique, as we do not work with time
series but with seasonal cycles. Also, usually the PCA/EOF are used to assess mode
of variability, so trends are removed. Here we want to describe the trend, but through
changes in regions having typical seasonal cycles. The two approaches are therefore
different. In our case, we don't detrend the data.
• L. 264: what does "normalizing by the correlation matrix" mean here? Divide
entry-wise by each element? If multiplying by the inverse of this matrix, then use
this phrase – one cannot "divide" by a matrix. Also in L. 256 we read "normalized
by the inverse of the correlation matrix" while L. 264 "we read "normalized by the
correlation matrix". Could the authors clarify this?
Thank you. We now say: " we do not normalize the distance by the inverse of the
covariance matrix (as usually done for the Mahalanobis distance) but by the inverse of
the correlation matrix"
• L. 373: please explain how M_cor is constructed (related to our other comment
on the corresponding figure).
M_cor is the correlation matrix calculated from all seasonal cycles (all years and all
grid points). We now say: "The correlation matrix is computed for all nonzero
seasonal cycles for the period 1979-2023 above 55 °N. It is calculated from the
matrix of shape (73, 1123710), having 1123710 value of SIC for 73 dates".
We have modified this part, as we no longer used the Mahalanobis for the calculation
of probability. We use the Euclidean distance for that. We still use the Mahalanobis
for the clustering though. We now say:"The Mahalanobis norm, deriving from a
symmetric operator, effectively rotates the original physical phase space (here., date
of the annual cycle) to align with the data's natural directions—linear combinations of
the physical time axis. This transformation allows centroid detection in a space that
reflects the intrinsic structure of the data. Therefore, using the Mahalanobis distance
helps the clustering algorithm to follow the direction of the correlation and capture
the elongated shapes of clusters. When calculating the probability to belong to one
cluster, we do not need to work with the data's natural directions, but rather work in
the original physical time space. Therefore we use Euclidean distance for the
calculation of probability and the Mahalanobis for the clustering."
• L. 379-382: So there the authors used fuzzy k-means but then in the previous
section, it was not fuzzy ? This is not very clear, and calls for a better justification
of the transition between these two sections. Also, is there an objective criterion
to prefer fuzzy k-means over crisp assignments? In general, how do we measure
whether the k-means did a good job or not (is it with the Silhouette metric, and if
so, what is a "good" clustering)?

We now say: "This means that we use a "fuzzy" k-means clustering where the assignment is soft (each data point can be a member of multiple clusters) in contrast to a hard or crisp assignment  (each data point is assigned to a single cluster; Jain et al., 2010)."

**Presentation**
• L. 73 : We think « sea-ice » with a dash should be used when it is used as an adjective; when used as a noun, it should be « sea ice »

Thank you. We have modified accordingly.

• L. 140-157: on the regionalization: it is not entirely clear why this paragraph is here. As we understand, no regionalization is required since the k-means method picks the optimal clusters which can then be used to defined physically-relevant regions. We would propose to move this paragraph to the discussion, since the regionalization is more an outcome of the work than a pre-requisite.

We have made it more clear that we have a double objective: "We determine Arctic regions based on statistically different sea-ice seasonal cycles, and describe Arctic changes through the time evolving borders of the regions characterized by these seasonal cycles."

• L. 158-172: the emphasis on regionalization for this paragraph (which follows another paragraph on regionalization) shifts somewhat the research question from the initial goal (understanding the physical regimes) to another goal (defining objective geographic boundaries). We would propose to focus the work on the former, and to discuss later in the text how the k-means can also be seen as a way to provide physically-based regions based on distinct sea ice dynamics. Of course, identifying spatial clusters and delimiting regions are two tasks that go hand in hand, but to us the main research question is not always clear.

We have made it more clear that we have a double objective: "we determine Arctic regions based on statistically different sea-ice seasonal cycles, and describe Arctic changes through the time evolving borders."

• L. 200 to 202: If 5-day mean shows similar results than for 1 day why not keeping 1 day as temporal resolution ? Could the authors precise their reasons for using 5-day mean instead of 1 day?

As the computation is quite long, we keep it to 5-days, as it gives similar results as daily data. We have added the following figure in the supplementary to demonstrate this.

[Figure]

Figure S1: Comparison between monthly (left), 5-day (middle) and daily temporal resolution (right). The Silhouette coefficient for a number of clusters from 2 to 6 (top row), the four types of seasonal cycles (middle row) and a map of the four labels (stable, stabilization, unstable, and destabilization) used to describe the evolution of Arctic clusters based on sea-ice seasonal cycles (bottom row). In the top row, the box extends from the first quartile (0.25) to the third quartile (0.75) of the Silhouette coefficient. The whiskers indicate the 1st and 99th percentiles. The green-dashed and orange-solid lines indicate the mean and median values, respectively.

• Fig. 9 is never cited

Fig 9 is now Fig 10. We now say: "Figure 10 illustrates how we define the stabilization and destabilization labels."

• On printed sheets of paper, the figures do not render very well.
We have improved the quality (by using eps instead of png in inkscape).

• Line 358 to 359. The transition from the section 3.1 to 3.2 is quite enigmatic.
Section 3.1 seems to deal with deterministic clustering where each point is
assigned to a cluster while section 3.2 deals with probabilistic clustering. We
would propose to put the explanation of lines 362-363 at the end of the section
3.1 to make a smoother transition

As suggested, we introduced a smoother transition for the sections by naming the
reason for introducing the probabilities: "As a given seasonal cycle can be in between
two or more seasonal cycle centroids, we introduce the probability to belong to one
cluster in the next section."

• Line 392: Could the authors explain why they chose to divide the period into 3
sub-periods of 15 years ? It seems a bit arbitrary and it's hard to understand
where the authors are going with this sub-time division. Could the authors add a
bit of context to introduce that choice?

We have 45 years of data.  45 is divided by 15 and 3 rows are a good compromise for
the size of the figures.

• Line 456 to 459 : We find it unclear whether the probabilities of belonging to a cluster given in this paragraph are the average of the probabilities over all the
years or just one year? Looking at Figure 7, it seems that this is for 1979. If that's
the case, why take one year rather than the average? Could the authors clarify
This?
Good point, we have now calculated the average and say: "The probability of
belonging to the open-ocean cluster is around 40%, to the permanent sea-ice cluster
is around 29% and to the full winter-freezing cluster is around 18 % and to the partial
winter-freezing cluster is around 13% (Figure 7)."
• Line 511. We are unsure about what the authors are trying to imply/measure by
defining "stable", "stabilization", "unstable" or "destabilization" regimes. In
particular, they should relate those regimes to something known from the
literature or give more interpretation because this notion and its application
come somewhat out of the blue.
We propose these new regimes to quantify the stability and transition of the Arctic
sea-ice seasonal cycles and refine the description in several parts of the text.
Minor comments
• L. 43 « optimal » should be accompanied by « (in the sense of statistical
dissimilarity) » or something like that, because otherwise this word can be
wrongly interpreted.
We now say: "Without providing prior information, this data-driven method shows
that the Arctic is best described by four types of seasonal cycles ..."
• L. 45. The use of the phrase "ice-free conditions" should be taken with care here
and throughout the manuscript, since ice-free has a well-defined meaning in the
climate projection literature (namely, sea ice extent < 1 million km2). What exactly
is ice-free in the context of this study? We also propose to add a horizontal line
on Fig. 4a to represent the chosen threshold, in order to quickly see when the
seasonal cycle shows ice-free conditions.
Thank you. We now clearly define that in this study ice-free conditions occur when
SIC < 0.15. To not overload the Figure 4a, we say it in the abstract :"two clusters
showing ice-free conditions (SIC < 0.15)"
and in the main text :"We refer to ice-free conditions when SIC is below 0.15."
• l. 43-45: When reading this part of the abstract, it sounds like the authors are re-
discovering two regimes that are well-known (open-ocean and permanent ice)
and two intermediate regimes where ice is present seasonally. Since the work
goes deeper than re-inventing these regimes, we would propose to describe the
clusters with more physical interpretation and to highlight what the k-means
analyses have allowed to do, that the human eye is not able to see (i.e., what is
the added value of an algorithm that can deal with large amounts of data in a high dimension space)

Thank you. We now say: "Without providing prior information, this data-driven method shows that the Arctic is best described by four types of seasonal cycles: …"

• l. 50 likelihood reduction à likely reduction? Or do the authors mean that the likelihood of this regime in the Canadian side has reduced? Maybe likelihood should be changed by "probability of occurrence" to make things clearer?

We now say: "The pan-Arctic probability to belong to the permanent sea-ice seasonal cycle has decreased by 3.1 %/decade which is compensated with an increase of probability to belong to the open-ocean cluster (1.6 % per decade), the full winter freezing cluster (1.1 % per decade) and to a smaller extent to the partial winter-freezing cluster (0.5 % per decade)"

• L. 52. You mean that spatial redistributions occur within the four clusters ?

Yes, exactly. We now say "spatial redistributions"

• L. 53-55 : it is grammatically strange to write that a "sea" is stabilizing. It is the state characterized by a given cluster that becomes less frequent, in a given region/sea.

We explicit what that means by saying: "From the Beaufort to the Kara Seas, the southern parts have stabilized (experiencing a new typical seasonal cycle, corresponding to the full winter-freezing cluster) and the northern part have destabilized (losing their typical permanent sea-ice seasonal cycle)."

• L. 54-55: be consistent grammatically: « have destabilized » … « have stabilized »…?

Thank you. We have modified it.

• L. 113 : % relative to what ?

It is relative to the period 1979 to 2018. We say: "the September SIE exhibits a decreasing trend of - 12.8 ± 2.3% per decade over the period 1979 to 2018".

• L. 116 : maybe « and do not consider changes in the underlying processes" ?

We want to emphasize that, on top of analyzing if there is sea-ice or not, changes can be made within the sea-ice regime. We say: "However, trends of SIA or SIE only inform about changes in regime from ice to open-ocean and do not consider changes in sea-ice features."

• L. 125: it would be worth mentioning that these studies have highlighted an asymmetry in the trend of retreat vs advance (see Lebrun et al. In particular).

Indeed, the current manuscript also reveals an asymmetry in the seasonal cycle, so there is a nice connection to be made here.

Exactly. We say that in the following paragraph: "(iii) the trend of later ice advance is expected to eventually double that of earlier retreat over this century, shifting the ice-free season into autumn (Lebrun et al., 2019)"

• L. 128 it is true that these studies do not inform on the sea ice dynamics including melt and growth behaviours, but does the present study do so, given that it does not seek to study the mass balance terms nor the time derivatives of sea ice Concentration?

The melt and growing behaviors are taken into account in our analysis through diagnostics of three different shapes of sea-ice. It includes the time derivative of SIC. But we agree that dynamics could not be fully accounted for without considering the thickness.

We add in the limitation:"Another limitation of this study is that sea-ice dynamics are analysed using sea-ice concentration, rather than sea-ice volume (which would better represent sea-ice behaviour, including growth and melting), due to the lack of robust and long-term sea-ice thickness data."

• L. 131-132. The statement that previous studies have not directly considered the full sea ice seasonal cycle, is not entirely correct. For example, previous studies looking at SIE report the seasonality of trends, of the mean state, of the variability. But they consider each point of the season separately, while in the present study the seasonality is accounted implicitly for while producing the clusters, through the correlation matrix.

Yes, previous studies have reported the seasonality of trends, which we state in the same sentence by saying: "These three ways of describing the variations in Arctic SIC (trend of SIE, type of sea-ice, ice-free duration), without considering directly the full sea-ice seasonal cycle, have nonetheless highlighted changes in the shape of the sea-ice seasonal cycle: (i) the trend in SIE depends on the season...". However, we think that these studies have not accounted directly for the full seasonal cycle in the diagnostics.
We also add in the limitation:"The major drawback of our approach resides in the exact grid point quantification of the real seasonal cycle features, as we gather grid cells within a type represented by a single seasonal cycle (the centroid). However, considering the full seasonal cycle gives useful information, as its derivative gives the period of melting and growth. Therefore, the two diagnostics complement each other nicely."

• L. 216 a "non-zero" seasonal cycle could be interpreted differently by different readers. We assume you mean "having at least a non-zero value for SIC throughout the year" ?

We now say: "We consider all oceanic grid cells above 55°N having a non-zero
sea-ice seasonal cycle (having at least a non-zero value for SIC throughout the year)"
• Line 298: "one of the two outputs" leads the reader to wonder what the other
output might be, and it is not displayed in the same paragraph. It is only
introduced at line 347. Please mention that the other output (the connection of
each grid point to a cluster) is studied later.
We have reorganized this paragraph to introduce directly the two outputs. We now
say: "The clustering method connects each seasonal cycle to a given cluster  (Figure
4a) and provides the centroids of each cluster (Figure 4b)."
• L. 302: follows à lags?
Yes, we now say: "follows the minimum solar insolation by a lag of around 3 months"
• L. 339 : apparition à appearance ?
We have corrected it. Thank you.
• Fig. 4 : the caption is not sufficient to understand what is plotted. Panel (a) shows
the average concentration over each cluster, is that correct? How is the time
dimension dealt with here? Are the average SIC determined for each cluster, then
averaged in time?
Panel a shows the centroid of the clustering method. These centroids are seasonal
cycles. It is the output of the clustering method.
We now say in the caption: "Four types of seasonal cycles (output of the clustering
method, called centroids)"
• Fig. 4b. have the authors plotted these maps for iconic years like 2007 (big ice
arch between the Greenland and the Kara Sea or 2012 (absolute minimum)? That
would be interesting to see to what extent the method captures the physics of
those events.
We now comment on these two years as follows: " Also, looking at the years with
marked extremes in September sea ice extent, (2007, 2012, 2016 and 2020; see
introduction), the MIZ categorization shows a transfer of area between the packed
ice and the MIZ. In our clustering vision, 2007, 2012 and 2020 show a transfer of
area between the permanent sea-ice cluster and full winter-freezing cluster while
2016 show a transfer of area between the full winter-freezing and the partial winter
freezing, reflecting different dynamical changes in the sea-ice seasonal cycles.
Therefore, our clustering analysis presents a more detailed description of the MIZ
category. "
• Line 370 centroids à centroid
We have corrected it. Thank you.

• Fig. 5: From a quick look, one could argue that the method is not really able to sort the data, as all four clusters have ~25% probability near the ice edge. We were both surprised to see so little variability in the maps, as they seem to have quite homogenous spatial distributions of probabilities.

Thank you very much. We have now decided to calculate the probability using the euclidean distance, which better splits the regions (we still use mahalanobis for the clustering).

We now say: "The Mahalanobis norm, deriving from a symmetric operator, effectively rotates the original physical phase space (here, date of the annual cycle) to align with the data's natural directions—linear combinations of the physical time axis. This transformation allows centroid detection in a space that reflects the intrinsic structure of the data. Therefore, using the Mahalanobis distance helps the clustering algorithm to follow the direction of the correlation and capture the elongated shapes of clusters. When calculating the probability to belong to one cluster, we do not need to work with the data's natural directions, but rather work in the original physical time space. Therefore we use Euclidean distance for the calculation of probability and the Mahalanobis for the clustering."

The new Figure

[Figure]

Figure 8: Map of the probability of each cluster: open-ocean (first column), partial winter-freezing (second column), full winter-freezing (third column) and permanent sea-ice (fourth column). Rows correspond to three periods of 15 years: 1979-1993 (top row), 1994-2008 (middle row) and 2009-2023 (bottom row). The dotted thin and thick lines are the mean SIC of 0.15 and 0.8 for the period 1979-2023, respectively. The circle sitting over the north pole is the pole hole (see section 2.1).

• Fig. 7. How is the "total probability" calculated here? Each spatial point has been assigned a probability, are these probabilities then averaged in space? Does it make physical sense to average probabilities in space?

The total probability is calculated by assigning to each spatial point, four probabilities (one for each cluster) and then summing for each cluster its associated probability over the whole domain. We do not average in space, we sum.

We say:" We call the total probability, Pt, the normalized area weighted probability over all grid cells. We sum, for each year, the probability weighted by the area of each grid cell over all grid cells divided by the sum of the probability weighted by the area of each grid cell over all clusters and all grid cells.

• Fig. 8: y-label units should be m2 not km2

Thank you very much. We have corrected it.

• Fig. 8: please repeat in the caption how Open-ocean, MIZ and Packed ice are Defined.

Good idea. For clarity, we now say in the caption: "Figure 8: (a) Time series of the total area covered by each of the four clusters. (b) Times series of the area covered by three categories: packed ice (0.8 < SIC < 1), the Marginal Ice Zone (MIZ; 0.15 < SIC < 0.8) and the open-ocean (SIC< 0.15).

• Fig. 8: how do we explain that there is much less interannual variability in the (b) panel (that is, hand-made clustering) compared to (a) (k-means)?

Thank you so much. Thanks to your comment we have noticed a bug when plotting the old Figure 8b. We have corrected it and the new figure has a similar interannual variability for panel a and b.

[Figure]

Figure 6: (a) Time series of the total area covered by each of the four clusters. (b) Times series of the area covered by three categories: packed ice (0.8 < SIC < 1), the Marginal Ice Zone (MIZ; 0.15 < SIC < 0.8) and the open-ocean (SIC< 0.15). All curves show a significant linear trend with a p-value less than 0.05 using a Wald Test with a t-distribution.

• Lines 505-509: it would be good to interpret the fact that 2007 is only visible for the permanent cluster, in terms of existing literature that studied this event. In fact, we would argue that the signature is also visible in the full winter-freezing cluster, in "negative". That is, the 2007 event seem to be interpretable as a strong drop of perennial ice but also as a strong surge of the full winter freezing cluster. Why is it so? We would have expected to see a surge of the open water cluster in 2007, not the winter cluster.

Thanks. We now say:"Also, looking at the years with marked extremes in September sea ice extent, (2007, 2012, 2016 and 2020; see introduction), the MIZ categorization shows a transfer of area between the packed ice and the MIZ. In our clustering vision, 2007, 2012 and 2020 show a transfer of area between the permanent sea-ice cluster and full winter-freezing cluster while 2016 show a transfer of area between the full winter-freezing and the partial winter freezing, reflecting different dynamical changes in the sea-ice seasonal cycles. Therefore, our clustering analysis presents a more detailed description of the MIZ category. "

Fučkar, N. S., Guemas, V., Johnson, N. C., Massonnet, F., & Doblas-Reyes, F. J. (2016). Clusters of interannual sea ice variability in the northern hemisphere. Climate Dynamics, 47(5), 1527–1543. https://doi.org/10.1007/s00382-015-2917-2

Lukovich, J. V., & Barber, D. G. (2007). On the spatiotemporal behavior of sea ice concentration anomalies in the Northern Hemisphere. Journal of Geophysical Research: Atmospheres, 112(D13). https://doi.org/10.1029/2006JD007836

Raphael, M. N., & Hobbs, W. (2014). The influence of the large-scale atmospheric circulation on Antarctic sea ice during ice advance and retreat seasons: RAPHAEL AND HOBBS; ANTARCTIC SEA ICE ADVANCE AND RETREAT. Geophysical Research Letters, 41(14), 5037–5045. https://doi.org/10.1002/2014GL060365

---

## Author Comment (AC3)

Marion Lebrun

**General Comments**

This paper introduces a novel method for characterizing the seasonality of Arctic sea ice. The k-means clustering method, which has been successfully applied in other contexts, serves as the foundation for this approach. This study enhances the method by incorporating the Mahalanobis distance, providing a more robust physical representation of sea ice seasonal cycles. By applying this refined technique to sea ice concentration data spanning 1979 to 2023, the authors identify four distinct clusters that effectively describe the seasonality of Arctic sea ice.

Building on these results, the authors analyze the probability of individual seasonal cycles within the dataset belonging to each of these four clusters. They also introduce new diagnostic tools to describe the temporal evolution of sea ice seasonal cycles, as well as to pinpoint the moments when a seasonal cycle transitions from one cluster to another.

This study offers an innovative approach that complements previous research on sea ice seasonality. I commend the authors for the thoroughness of their analysis and the clarity of their well-structured manuscript. Overall, I believe this work represents a significant contribution to the field and, with the revisions suggested below, has the potential for a strong impact upon publication.

While the bibliography section requires attention before submission (as detailed later), most of the comments below are intended to enhance the paper's clarity, precision, and relevance. Authors are encouraged to consider these suggestions and apply them at their discretion.

**Major Comments**

The only major comment regarding this paper concerns the paragraph between lines 325 and 339. In my opinion, this paragraph requires further clarification or a more detailed presentation of the results to be fully convincing. The results presented in this section and reiterated in the conclusion, showing that ice conditions during the summer (or winter) are strongly correlated with the onset of melting (or freezing), are highly interesting for understanding the mechanisms behind the Arctic sea ice seasonal cycle. However, I find that relying solely on clusters to describe these results is limiting.

If I understand correctly, the seasonal cycles of each grid point tend to group around one (or several, considering the results in Section 3.2) cluster. Therefore, I am not fully convinced that the seasonal cycles associated with a particular cluster (as shown in Figure 4b) behave exactly as summarized in this paragraph.

I believe it would be helpful to visually demonstrate this with supporting evidence to strengthen the argument. For example, you could present the interquartile range around each cluster on Figure 4a, using the data already employed to generate Figure

4b. Alternatively, if you prefer not to overload Figure 4a, you could clearly define the
melting onset and freezing onset dates here (using concentration thresholds already
applied in other studies) and provide statistics of these diagnostics in each cluster. In
my opinion, these revisions would significantly enhance the impact and clarity of
these results.

Thank you very much for your relevant comment and nice idea. We have further
pushed the diagnostic.

To have an idea of the spread of seasonal cycles around each cluster, we have also
plotted the median (solid line) together with quantiles 0.90 and 0.10 (dashed line) for
each cluster in a supplementary figure.

[Figure]

Figure S2: As Fig. 4a, but for the median (solid line) and quantiles 0.10 and 0.90
(dashed line)

In the main text, we have added a new figure and interpretation considering the
spread for the date of retreat and date of advance.

We now say:

"... Therefore, it seems that, for ice-free conditions in summer, the first date of
freezing is a good predictor for the appearance of full ice conditions in the next
winter.

However, this suggestion relies solely on the shape of the four types of
seasonal cycles but to properly quantify this, the spread must be taken into account.
Figure S2 displays the spread of the seasonal cycle by plotting the quantiles 0.1, 0.5
and 0.9 of each cluster. To verify our hypothesis on sea-ice predictors, we account for
the spread of the date of retreat and date of advance for each cluster. To do so, we

calculate the first date of retreat (the first date after the maximum SIC that is below 0.9) for each seasonal cycle experiencing fully ice covered conditions (having at least one value above 0.99 during the year). We also calculate the first date of advance (the first date after the minimum SIC that is above 0.1) for each seasonal cycle experiencing ice-free conditions (having at least one value below 0.01 during the year). For these calculations, seasonal cycles have been temporally filtered using a 15 days sliding window in order to get rid of short-term dynamical ice events, as done in Lebrun et al., (2019). To circumvent the effect of the discontinuity between 31 December and 1 January, we define the origin of time in May for the calculation of the date of advance. We then label each first date of retreat and first date of advance for each seasonal cycle with its corresponding cluster according to our clustering analysis (Figure 4a).

The normalized probability over each cluster of the first date of retreat and first date of advance at each date is shown Figure 5. This figure also displays the total number of the first date of retreat and the first date of advance of all clusters for each date. If the first date of retreat occurs between January and April, there is around 95% of chance to belong to either the open-ocean cluster, the partial winter-freezing cluster or full winter freezing cluster, which all present ice-free duration in the following summer. However, this situation did not often occur, as the total first date of retreat happening in this period is unlikely (solely 5% of first date of retreat for all clusters). The first date of retreat is more likely to occur between the beginning of April and August, as within this period around 90% of the total date of retreat for all clusters exist. A first date of retreats in early June has solely around 10% of chance to belong to the permanent sea-ice cluster which do not present ice-free conditions in summer while a first date of retreat in early July has around 70% of chance to belong to the full winter-freezing cluster which shows ice-free conditions in summer.

The first date of advance is more likely to occur between the beginning of August until the beginning of January, as within this period around 90% of the total date of advance for all clusters exist. A first date of advance in early September has around 95% of chance to belong to the full winter freezing cluster which present fully ice covered condition in the following winter, while a first date of advance in early

November has around 80% of chance to belong to the partial winter freezing or open ocean clusters which do not show fully ice covered conditions in the following winter. Therefore, this simple model suggests that the first date of retreat could be a good predictor for ice-free conditions the following summer and the first date of advance a good predictor for fully ice cover conditions the following winter. "

[Figure]

Figure 5: Normalized probability of the first date of retreat (panel a) and first date of advance (panel b) for each cluster. The solid lines with star markers are the total number of first dates of retreat and first dates of advance for each date. The green circle markers (start date) and green square markers (end date) cover the shortest period where around 90% of the first date of retreat, respectively the first date of advance,  for all clusters occurs.

**Specific Comments**

**Lines 137-139 and lines 152-157:**  I feel, with both sentences, that the authors try to highlight the novelty of their method compared to previous studies, but these sentences appear before the work itself is introduced.  I am not sure of the relevance of these sentences at this point. This creates some confusion during a first reading. I suggest either removing these sentences or moving them to a later paragraph, ideally after the authors have introduced their work more clearly.

The lines 137-139 *("Here, in this paper, we describe the evolution of the Arctic by delimiting spatio-temporal regions having a common type of seasonal cycle.")*  and the lines 152-157 *("However, the criteria for the boundaries of these proposed regions are hard to determine and somewhat arbitrary. The originality of our analysis also resides in the fact that we regionalize the Arctic based on physical criteria of the dynamics of the sea-ice seasonal cycle, therefore without imposing predefined regions. To do so, we set up a clustering method (unsupervised machine learning")* and started  introducing our work in regards to previous work. We think by removing or moving to the last paragraph, the originality of our work will be less easy to follow. In this manner, we can easily and step by step compare the differences/added value of our work to previous studies.

**Lines 200-202:** This statement could benefit from additional  evidence (by including figures or statistics in the supplementary material, for example).

We agree that the lines 200-202 *('We choose this 5-day temporal resolution as similar results are found for a daily temporal resolution whereas a monthly temporal resolution shows small differences in the spatial distribution of clusters.)*  could benefit from additional evidence.

We now say: "We choose this 5-day temporal resolution as similar results are found for a daily temporal resolution whereas a monthly temporal resolution shows different numbers of optimal clustering and small differences in the spatial distribution of clusters (Figure S1)."

We have included this new figure and associated paragraph in the supplementary

[Figure]

Figure S1: Comparison between monthly (left), 5-day (middle) and daily temporal resolution (right). The Silhouette coefficient for a number of clusters from 2 to 6 (top row), the four types of seasonal cycles (bottom row). In the top row, the box extends from the first quartile (0.25) to the third quartile (0.75) of the Silhouette coefficient. The whiskers indicate the 1st and 99th percentiles. The green-dashed and orange-solid lines indicate the mean and median values, respectively.

The Silhouette coefficient is maximum (so optimal clustering) with 6 clusters with monthly data and for 3 clusters for 5-days and daily data.

**Lines 216-217:** It could help the reader if you briefly explain what you mean by "*non-zero seasonal cycle*."

We now say: "We consider all oceanic grid cells above 55°N having a non-zero sea-ice seasonal cycle (having at least a non-zero value for SIC throughout the year)"

**Lines 353:** "*consistent and continuous patterns*", consistent according to what?

We have removed "consistent". We now say: "we retrieve spatially continuous patterns"

**Lines 395-397:** I find it difficult to discern "*the edge of the 0.3 probability*" in Figure 5 due to the continuous colorbar. Adding a contour line to indicate this boundary could make it clearer.

We think it will unnecessarily overload the plots. We now say: "The edge of the 0.3 probability of belonging to the permanent sea-ice clusters of the period 1979-1993 follows the border of the Marginal Ice Zone (0.8 SIC) located in the Central Arctic (not shown)"

**Figure 5:** I find it difficult to discern the probability differences between clusters when the colorbar changes for each cluster. While I understand your choice to maintain consistency with other figures, for this particular figure, I suggest using a single colorbar for all four clusters to enhance readability.

The figure now has clear separation between probabilities, as we no longer used the Mahalanobis for the calculation of probability. We use the Euclidean distance for that. We still use the Mahalanobis for the clustering though. We now say:"The

Mahalanobis norm, deriving from a symmetric operator, effectively rotates the original physical phase space (here., date of the annual cycle) to align with the data's natural directions—linear combinations of the physical time axis. This transformation allows centroid detection in a space that reflects the intrinsic structure of the data. Therefore, using the Mahalanobis distance helps the clustering algorithm to follow the direction of the correlation and capture the elongated shapes of clusters. When calculating the probability to belong to one cluster, we do not need to work with the data's natural directions, but rather work in the original physical time space. Therefore we use Euclidean distance for the calculation of probability and the Mahalanobis for the clustering."

The new figure : "

[Figure]

Figure 8: Map of the probability of each cluster: open-ocean (first column), partial winter-freezing (second column), full winter-freezing (third column) and permanent sea-ice (fourth column). Rows correspond to three periods of 15 years: 1979-1993 (top row), 1994-2008 (middle row) and 2009-2023 (bottom row). The dotted thin and thick lines are the mean SIC of 0.15 and 0.8 for the period 1979-2023, respectively. The circle sitting over the north pole is the pole hole (see section 2.1). "

**Paragraph 464-476:** The reference to Figure 7 is missing here and should be explicitly mentioned for clarity.

Thank you. We now say: " The probability of belonging to the open-ocean cluster is

around 40%, to the permanent sea-ice cluster is around 29% and to the full winter-freezing cluster is around 18 % and to the partial winter-freezing cluster is around 13% (Figure 7)"

**Line 468:** " *the trend for the other two clusters are statistically significant*". Does it mean that the trends for the partial and full winter-freezing clusters are not significant?

Yes, we have now clarified by saying:"All curves show a significant linear trend with a p-value less than 0.05 using a Wald Test with a t-distribution."

**Lines 476:** "*and to a smaller extent, of the full winter-freezing cluster*". I am confused by this statement. Earlier (line 466), it was mentioned that the trend for the full winter-freezing cluster was nearly constant, and line 468 suggests that the trend is not significant. This part seems to lack precision to be clearly understood.

Indeed, it was confusing. We have removed this last part of the sentence.

**Lines 502-503:** "*while the partial and full winter-freezing clusters remain relatively stable*." I find it unclear in Figure 8a that the total area covered by the partial and, in particular, the full-winter cluster is stable. It might be helpful to add the trend for each cluster, as was done in Figure 7, to make this clearer.

Thank you very much. We have now computed the trend and significance.

[Figure]

Figure 6: (a) Time series of the total area covered by each of the four clusters. (b) Times series of the area covered by three categories: packed ice (0.8 < SIC < 1), the Marginal Ice Zone (MIZ; 0.15 < SIC < 0.8) and the open-ocean (SIC< 0.15). All curves show a significant linear trend with a p-value less than 0.05 using a Wald Test with a t-distribution.

And we now say:"These two methods (Figure 6a and Figure 6b) both indicate a shift toward more seasonal and ice-free conditions. Indeed, in the clustering method the permanent sea-ice cluster has notably decreased of the same amount than the packed ice in the classical categorization (-0.8 .10⁶km² per decade). Also, the open-ocean cluster follows the same trend of the open-ocean category (0.3 10⁶ km² per decade). The increase in the area of MIZ category is around 0.5 10⁶ km² per decade and has been demonstrated previously (Cocetta et al., 2024; Song et al., 2025). Therefore, it appears with our clustering that the MIZ is refined into two clusters : the full winter-freezing (0.3 10⁶ km² per decade) and the partial

winter-freezing cluster (0.2 10⁶ km² per decade). This suggests that the tendency is more likely to shift to a more abrupt melting and growth seasonal cycle (full winter-freezing cluster) compared to a quasi-sinusoidal sea-ice seasonal cycle (partial winter-freezing cluster). "

**Lines 505-509:** I find it difficult to discern the nuances between these two sentences. Perhaps the last sentence could be omitted, as it might not add significant value.

We agree. We have removed the last sentence.

**Lines 518-519:** "*Sensitivity tests have been performed on this definition, and the results do not change when we apply small definition changes (i.e., 9 to 11 years minimum length of the same cluster with zero to 2 years of tolerance).* " It would be valuable to include the results of these sensitivity tests, perhaps in the supplementary material, to provide additional context and support for this statement.

We added the following figure in the supplementary and we now say: "Sensitivity tests have been performed on this definition (Figure S2)"

[Figure]

Figure S3: Sensitivity tests on our definition of regime. Same as Figure 11 but with a different set of values for the minimum number of consecutive years and tolerance.

**Lines 522-525:** A reference to Figure 9 would enhance clarity here, as the figure significantly helps in understanding the definition of these new diagnostics.

Thanks. Figure 9 is now Figure 10. We now say: "Figure 10 illustrates how we define the stabilization and destabilization labels."

**Lines 578-582:** I am uncertain about the relevance of this paragraph, as all the information presented here seems to be already covered in the previous paragraph.

We agree that the Figure 11d is more a confirmation from the previous paragraph. However the Figure 11c does as it allows additional spatial information to which seasonal cycle the regions stabilized and informs that it depends on the regions. We keep both figures for consistency among the stabilization and destabilization results. We have reformulated the paragraph.

We now say: "To describe the stabilization and destabilization regimes, we display the dominant cluster (the cluster having the maximum probability) during the stable phase of these two regimes (early period for the destabilization regime and late period for the stabilization period; Figure 12c and 12d). "

**References section :**

I noticed that several references cited in the paper are missing from the reference list. I've compiled a list of the missing references I found, but I strongly recommend that the author carefully review this section, as there may be other errors that I may have missed. Additionally, the format of the references is not consistent throughout the section. For example, the publication date is sometimes listed immediately after the authors' names, and in other cases, it appears at the end of the reference. To ensure consistency, I suggest following the EGU standardized citation format.

Thank you very much. We have now completed the missing references and ensure that the year is right after the name's author in a consistent way.

**Missing reference in the references section :**

Ardyna and Arrigo, 2020: line 83

Eisenman, 2010: line 439 ; 693

Eyring et al., 2021: line 77

Lebrun et al., 2019: line 125 ; 137 ; 631 ; 648

Meier et al., 2007: line 146

Markus et al., 2009: line 127 ; 629

Maze et al., 2017: line 161

Parkinson and Cavalieri, 2008: line 678

Parkinson et al., 1987:  line 304

Parkinson et al., 1999: line 677

Parkinson, 2014: line 127

Peixoto and Oort, 1992: line 683

Peng and Meier, 2018: line 146

Regan et al., 2023: line 119

**Lines 803-805:** Houghton and Wilson 2020 should appears before Huntington et al., 2017

---

## Author Response (AR2)

**ROUND 2**

1.  Provide an algorithm flowchart / schematic
    We have added a schematic Figure 2c, panel to specify our data input for the k-mean clustering.

[Figure]

Figure 2: Schematic of the matrix input data for the k-means clustering (panel a), correlation matrix of the 5-day mean SIC for all non-zero sea-ice seasonal cycle above 55°N (panel b) and algorithm flow chart of the clustering (panel c)

2.  Provide code on Github or equivalent
    We have now provided the code. We now say in the methodology section: "The code developed for this study is available for download at https://github.com/amelie-simon-pro/SIC_Clustering."

    And in the code and data availability section: "The code developed for this study

is available for download at https://github.com/amelie-simon-pro/SIC_Clustering. We utilized Mistral (https://chat.mistral.ai/chat) and ChatGPT (https://chat.openai.com/) to assist in generating small portions of the code, which we subsequently adapted for our script."

3. Improve equation formatting that appear copied and pasted
   We use Equation Editor in Google docs and tried to export in pdf using different computers. If the quality is not sufficient, we will take advice at the publication production process stage.

4. Clarify how silhouette coefficients are helpful in selected number of clusters

   We have modified the text and now say: "Clustering methods are a type of unsupervised learning technique where the number of underlying classes, called clusters, is unknown beforehand. Consequently, one must test several choices for the number of clusters, k. For each chosen value of k, a metric must be calculated to evaluate the resulting partition. The Silhouette coefficient is a metric classically used for this purpose (Rousseeuw, 1987; Houghton and Wilson, 2020). The general idea is to measure how similar an object is to its own cluster compared to other clusters; a high Silhouette value means the object is well matched to its own cluster and poorly matched to neighboring clusters. Indeed, the larger the Silhouette coefficient is (bounded between -1 to 1), the farthest the centroids are from each other and the more grouped are the points of the same cluster. It measures the quality of the clustering when seeking for compact and well-separated clusters. Ultimately, the number of clusters that maximizes the Silhouette coefficient is the optimal choice retained for the final clustering solution. We rely on the Silhouette_sample function from the python package sklearn.metrics (Pedregosa et al., 2011), which calculates the Silhouette coefficient for every point as (b - a) / max(a, b) where $a$ is the mean intra-cluster distance and $b$ is the mean distance for each point to its neighbouring cluster (closest cluster for which the point is not being part)."

5. Expand a little bit more why the new classes of partial and full winter freezing are helpful for sea ice process understanding and for forecasting applications. Can you quantify this somehow?
   We further highlight the added-value of our description compared to the MIZ (Figure 6).

   We now added in the abstract: "The new classes of partial and full winter freezing are helpful for sea ice process understanding as it refines the classical MIZ category into two distinct sea-ice clusters. The trend is primarily controlled by the tendency of  the more abrupt melting and growth seasonal cycle (full winter-freezing cluster) compared to the trend of the  quasi-sinusoidal sea-ice seasonal cycle (partial winter-freezing cluster)."

   We now say in the discussion: "The added value of our description compared to

the MIZ category (sea-ice concentration between 0.15 and 0.8) is the new classes of partial and full winter freezing. This could be important for sea-ice dynamics and forecasting understanding. Our result suggests that the trend is primarily controlled by the trend of the more abrupt melting and growth seasonal cycle (full winter-freezing cluster) compared to the trend of the quasi-sinusoidal sea-ice seasonal cycle (partial winter-freezing cluster) or, in other words, that the trend is more likely due to increase of regions with total ice cover in winter with a short ice-free season (2 months, full winter-freezing cluster) than increase of regions with a partial ice cover in winter with a long ice-free season (4 months, partial winter freezing cluster)."

And:
"These research findings which are relevant for climate models and process understanding, can also provide useful information for forecast application, guiding ecosystem conservation efforts, and thus related policy-making planning."

6. Are your results sensitive to the time span analysed?
   Thank you. We now say: "Sensibility tests suggest that 45 years of data are long enough to obtain a robust signal, as close clusters are obtained using periods of 20 years, 30 years and 40 years (Figure S2)."

[Figure]

Figure S2: Same as Fig. 4b but for different time periods: 20 years (1979 to 1998, panel a; 2004 to 2023, panel b), 30 years (1979 to 2008, panel c), and 40 years (1979 to 2018, panel d)